# TimeRAF: Retrieval-Augmented Foundation model for Zero-shot Time Series Forecasting

## Abstract

Time series forecasting plays a crucial role in data mining, driving rapid advancements across numerous industries. With the emergence of large models, time series foundation models (TSFMs) have exhibited remarkable generalization capabilities, such as zero-shot learning, through large-scale pre-training. Meanwhile, Retrieval-Augmented Generation (RAG) methods have been widely employed to enhance the performance of foundation models on unseen data, allowing models to access to external knowledge. In this paper, we introduce **TimeRAF**, a **R**etrieval-**A**ugmented **F**orecasting model that enhance zero-shot time series forecasting through retrieval-augmented techniques. We develop customized time series knowledge bases that are tailored to the specific forecasting tasks. TimeRAF employs an end-to-end learnable retriever to extract valuable information from the knowledge base. Additionally, we propose Channel Prompting for knowledge integration, which effectively extracts relevant information from the retrieved knowledge along the channel dimension. Extensive experiments demonstrate the effectiveness of our model, showing significant improvement across various domains and datasets.

## 1 Introduction

Time series (TS) forecasting has gained significant popularity in recent years due to its vital role in various domains, including finance (Yu et al., 2023), healthcare (Li et al., 2024), weather (Wu et al., 2023b), and traffic (Jin et al., 2021). The popular approach in the past typically learns from single-domain, small-scale datasets (Nie et al., 2023; Zeng et al., 2023), which inherently constrains their generalization capabilities. However, the landscape of time series analysis is evolving rapidly with the advent of large models. Time series foundation models (TSFMs), trained on large-scale, multi-domain datasets, have demonstrated zero-shot learning abilities, revolutionizing various time series domains and diverse applications (Liang et al., 2024; Woo et al., 2024; Liu et al., 2024).

Meanwhile, Retrieval-Augmented Generation (RAG) is an increasingly prevalent technique that enhances the capabilities of foundation models in various domains, including text generation (Karpukhin et al., 2020) and image generation (Chen et al., 2023). This approach allows models to access external knowledge through various information retrieval techniques, enabling them to gather supplementary information during the generation process. Typically, the retrieval knowledge can be sourced from external datasets in the same format with the training corpus. For instance, in dialogue systems, RAG can help generate more contextually relevant responses by retrieving previous dialogues or similar interactions from a database (Huang et al., 2023). However, similar studies have garnered little attention in the time series domain. A natural question arises: ***Can the integration of time series foundation models with retrieval-augmented methods also improve performance, particularly in challenging scenarios that require strong generalization abilities, such as zero-shot forecasting?***

As an intuitive example, a pre-trained model trained on a general time series dataset may struggle when forecasting for specific domains such as weather patterns in a particular region, which is illustrated in the left plot of Figure 1. However, by accessing domain-specific external knowledge bases, the model could dynamically retrieve relevant information—such as time series data from similar weather conditions—without requiring extensive parameter updates. This allows the model to integrate domain-specific prior knowledge, improving its zero-shot forecasting capability. In this manner, retrieved external data provides valuable context and serves as an additional source of prior information, enabling more accurate predictions. These advancements motivate our exploration of

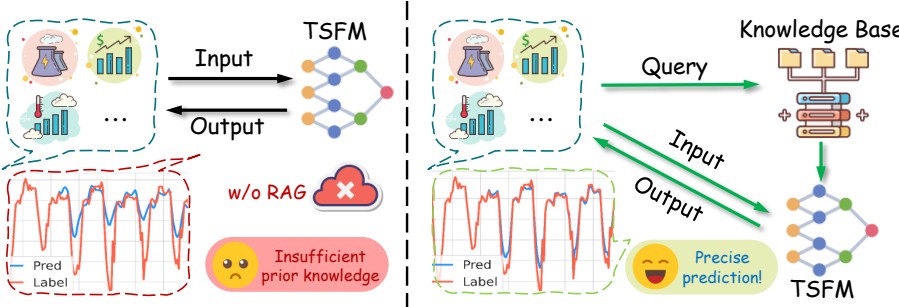

Figure 1: **Left:** Time series foundation models (TSFMs), while capable of zero-shot forecasting, are limited by insufficient prior knowledge, resulting in constrained prediction accuracy. **Right:** By dynamically retrieving relevant information from an external knowledge base, our TimeRAF enhances prediction accuracy, leading to more precise zero-shot forecasting performance.

**R**etrieval-**A**ugmentation for time series **F**orecasting (RAF). However, designing an effective RAF framework for time series forecasting involves several key challenges: **(1)** *What types of data can serve as knowledge bases to support time series models?* **(2)** *How can relevant knowledge be retrieved when encountering inputs from diverse domains?* **(3)** *How can retrieved knowledge be effectively integrated to improve model performance?*

To address these challenges, we introduce TimeRAF, a novel framework designed to leverage retrieval-augmented generation techniques for time series foundation models. As shown in the right of Figure 1, by retrieving and integrating external time series data, we aim to overcome the limitations of existing TSFMs and enhance zero-shot time series forecasting performance. TimeRAF consists of a retriever that scores and selects relevant time series data from an external knowledge base. The knowledge base can either be a comprehensive database composed of multiple datasets across various domains or a domain-specific database comprising a singular dataset relevant to test data. Furthermore, an end-to-end learnable retrieval methodology is introduced to ensure that the retrieved data delivers enhancement. To leverage retrieved time series, we introduce an effective approach, named Channel Prompting, to integrate the knowledge from retrieved data. Our extensive experiments on various datasets demonstrate that TimeRAF significantly achieves a substantial improvement over TSFM and outperforms several existing zero-shot time series forecasting methods.

Overall, our contributions can be summarized as follows:

- We propose TimeRAF, a novel framework that leverages retrieval augmentation techniques to enhance zero-shot time series forecasting. By retrieving relevant data from an external knowledge base and effectively integrating the retrieved information, TimeRAF supplements the pre-trained knowledge of foundation models, enhancing their forecasting capabilities.

- We employ a learnable retriever to calculate retrieval scores for time series within the knowledge base and select the best options. To integrate retrieved knowledge, we introduce Channel Prompting to extract valuable information from the retrieved data effectively.

- Our TimeRAF demonstrates significant improvement through the incorporation of RAF into TSFM and even outperforms several full-shot methods. Furthermore, we present comprehensive ablation studies and visualizations to evaluate the efficacy of our approach.

## 2 RELATED WORK

**Foundation Models for Zero-shot Time Series Forecasting:** Recent years have witnessed the rise of TSFMs. TimeGPT-1 (Garza & Mergenthaler-Canseco, 2023) is the first closed-source model offering zero-shot forecasting capabilities. ForecastPFN (Dooley et al., 2024), pre-trained on synthetic time series data, serves as a zero-shot forecaster, but excels primarily in data- or time-limited settings. Lag-llama (Rasul et al., 2023) leveraged the LLaMA architecture (Touvron et al., 2023) with lagged time series features for time series forecasting. TimesFM (Das et al., 2024)is a patch-based, decoder-only foundational model designed for time series forecasting, which employs a larger output patch size to enhance decoding efficiency. The model is pre-trained on a comprehensive dataset sourced

from Google Trends and Wikipedia pageviews, in combination with open data. MOIRAI (Woo et al., 2024) introduces LOTSA, a large-scale collection of open time series datasets, and utilizes it to train a foundation model based on a masked encoder architecture. MOIRAI achieves competitive or superior performance as a zero-shot forecaster when compared to full-shot models. Tiny Time Mixers (TTMs) (Ekambaram et al., 2024) leverages a lightweight mixer-style architecture and demonstrated remarkable zero-shot forecasting performance. Since TSFMs have shown potential in zero-shot time series forecasting, our approach aims to enha nce their generalization capabilities by applying RAG techniques to leverage external knowledge.

**Retrieval Augmented Generation for Foundation Models:** Foundation Models like LLMs have achieved remarkable success, though they still face limitations in domain-specific or knowledge-intensive tasks. To address these challenges, various RAG methods have been proposed: DocPrompting (Zhou et al., 2022) curated a retrieval annotation dataset to train a retriever for augmenting input in code generation. DPR (Karpukhin et al., 2020) develops a dense embedding model for indexing passages in a low-dimensional, continuous space. RePlug (Shi et al., 2023) refined the retriever by distilling the knowledge from the language model's probability. LAPDOG (Huang et al., 2023) introduces an end-to-end dense retriever framework specifically for personalized dialogue generation, emphasizing objective optimization. Beyond NLP tasks, RAG has also been applied to other domains: REACT (Liu et al., 2023) freezes the original model and updates only the additional trainable weights on the retrieved knowledge, significantly enhancing visual model's zero-shot performance. Re-Imagen (Chen et al., 2023) uses retrieved information to produce high-fidelity and faithful images, even for rare or unseen entities. Additionally, in time series analysis, RATSF (Wang & Cui, 2024) develops a cross-attention module to integrate additional data for better prediction. But its retrieval process is constrained to historical data. ReTime (Jing et al., 2022) retrieves relational references to improve forecasting and imputation for incomplete time series. However, existing methods cannot adapt to zero-shot tasks. In contrast, our work is specifically designed to address this gap. We use extensive public time series data to build a knowledge base and enhance zero-shot prediction in TSFMs with an effective RAF method.

## 3 METHOD

### 3.1 OVERVIEW

An illustration of our TimeRAF framework is provided in Figure 2. Firstly, a retriever is utilized for learning to retrieve relevant data from the external knowledge base (refer to section 3.5). Following this, the proposed Channel Prompting approach is employed for the integration of retrieved knowledge. Therefore, the entire forecaster $\mathcal{F}$ is capable of harnessing external knowledge, thereby facilitating knowledge enhanced forecasting (refer to section 3.4). During training, the backbone of TSFM remains frozen. Details of training and inference process are provided in section 3.5.2 and section 3.6. Besides, the knowledge bases utilized for training and inference are detailed in section 3.3.

### 3.2 PROBLEM FORMULATION

Following previous work (Nie et al., 2023), we employ the channel inpendent strategy. Let $\boldsymbol{X} \in \mathbb{R}^{sl \times c}$ be a multivariate time series of length $sl$ and number of channels $c$. The input can be denoted as $\boldsymbol{x} \in \mathbb{R}^{sl \times 1}$, and the forecasting task can be formally defined as predicting the future values $\hat{\boldsymbol{y}} \in \mathbb{R}^{fl \times 1}$ given the history/lookback window $\boldsymbol{x}$. Upon completing the analysis of all data channels, the final comprehensive prediction result $\hat{\boldsymbol{Y}} \in \mathbb{R}^{fl \times c}$ is derived. Here, $fl$ denotes the forecast length/horizon. The ground truth is denoted as $\boldsymbol{Y} \in \mathbb{R}^{fl \times c}$. In the zero-shot forecasting setting, the model generates predictions for future values based on datasets that have not been encountered during training. Given a set of retrieved time series data $\boldsymbol{C}$ from the external knowledge base (details will be elaborated in Section 3.5), we aim to leverage the valuable information within them to enhance the forecasting capability of the forecaster $\mathcal{F}$. The entire process can be formulated as $\hat{\boldsymbol{Y}} = \mathcal{F}(\boldsymbol{X}, \boldsymbol{C})$.

### 3.3 KNOWLEDGE BASE

In order to facilitate knowledge retrieval, it is essential to first establish a knowledge base. To enhance the efficiency of knowledge integration and extraction, we undertake a preprocessing of all sequences

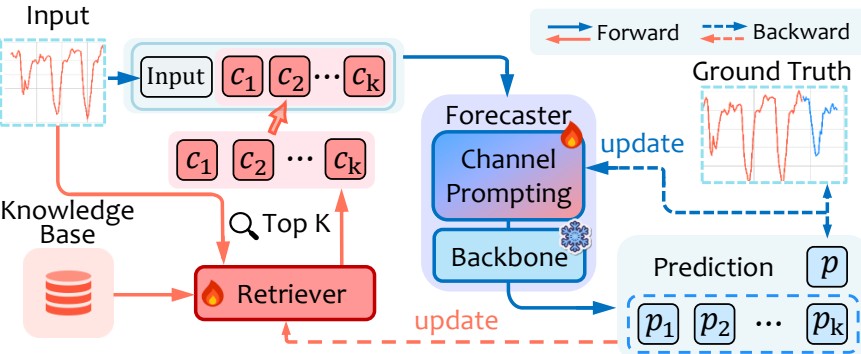

Figure 2: **Overview of TimeRAF:** TimeRAF utilizes a retriever to dynamically retrieve relevant candidates from an external knowledge base and then utilizes the proposed Channel Prompting module to integrate knowledge between the retrieved data and the input. The knowledge-enhanced embeddings are subsequently fed into the backbone of the foundation model to improve forecasting results. During training, the backbone remains frozen.

within the knowledge base to align with the dimensions of the lookback window, resulting in the following representation: Knowledge Base $= \{t_i | t_i \in \mathbb{R}^{sl \times 1}\}_{i=1}^{n_{kb}}$, where $n_{kb}$ represents the size of knowledge base. The data in the knowledge base will use the same normalization as the input. To maintain the generalization capabilities of the foundation model, we use multi-domain datasets for training, similar to the pre-training phase of the foundation model, which will be detailed in section 4.1.

Subsequently, we apply a sliding window with the same window size as the input of the foundation model across the training datasets. Based on the scale of each sub-dataset, we ultimately establish a knowledge base in which each domain has an equal proportion to uphold the balance. Additionally, there is no overlap present in the data within the knowledge base. During Training, to prevent data leakage caused by accessing future sequences, the retriever is constrained to retrieve information solely from datasets that are different from the dataset of input sequence. During inference, TimeRAF has the option to utilize the extensive multi-domain knowledge base that we have developed or to opt for a domain-specific dataset as the knowledge base, based on the specific requirements.

## 3.4 KNOWLEDGE INTEGRATION

Given $k$ retrieved time series data $\boldsymbol{C} = \{\boldsymbol{c}_1, \boldsymbol{c}_2, \ldots, \boldsymbol{c}_k\}$ from the external knowledge base (details will be discussed in Section 3.5), we aim to leverage the valuable information within them, thereby complementing the pre-trained knowledge of TSFMs to enhance forecasting performance.

Following the preprocessing procedure of the TSFM, each sequence $\boldsymbol{c}_i$ in $\boldsymbol{C}$ will undergo normalization followed by patching, analogous to the input. Thereafter, these patches will be processed through a projection layer to derive their respective embeddings. Let $\widetilde{\boldsymbol{x}} \in \mathbb{R}^{n \times d}$ denotes the input embedding and $\widetilde{\boldsymbol{c}}_i \in \mathbb{R}^{n \times d}$ represents the embedding of the $i$th retrieved candidate. Here, $n$ denotes to the number of patches, while $d$ indicates the dimensionality of the embedding.

The Channel Prompting begins with a flatten operation on both embedding of input and retrieved candidates. Subsequently, the flattened embedding of input and retrieved candidate will be concatenated:

$$\boldsymbol{z}_i = \text{Concat}(\text{Flatten}(\widetilde{\boldsymbol{x}}), \text{Flatten}(\widetilde{\boldsymbol{c}}_i)). \tag{1}$$

By integrating the input embedding with the external knowledge embedding, the representation of the input is enriched with supplementary contextual information. Furthermore, after obtaining the concatenated embedding $\boldsymbol{z}_i \in \mathbb{R}^{2*n*d}$, the foundation model is better positioned to comprehend the lookback window through the incorporation of domain-specific knowledge or external facts.

Subsequently, we employ a MLP to effectively extract and combine the most relevant information from both the lookback window and the retrieved candidates. This process enables the compression of

the combined representation into a more meaningful and compact form. In particular, the concatenated embedding $z$ is compressed back to the original dimensions corresponding to the foundation model, yielding $\widetilde{z} \in \mathbb{R}^{n \times d}$. Besides, the original input embedding is reintroduced through a residual connection to ensure the complete preservation of the information from the lookback window. The entire process can be formulated as follows:

$$\widetilde{x}^* = \widetilde{x} + \widetilde{z} = \widetilde{x} + \text{MLP}(z). \tag{2}$$

In the case where $k$ candidates are retrieved, each candidate will undergo the aforementioned processing steps, resulting in $k$ embeddings denoted as $\widetilde{z} = \{\widetilde{z_1}, \widetilde{z_2}, \ldots, \widetilde{z_k}\}$. Prior to implementing the residual connection, we compute the average of these embeddings $\widetilde{z}$, i.e. $\widetilde{x}^* = \widetilde{x} + \text{Avg}(\text{MLP}(z_1), \ldots, \text{MLP}(z_k))$. By extracting the relevant feature of the concatenated embedding, the foundation model is empowered to to incorporate additional contextual information from the external knowledge base. Consequently, the final knowledge-enhanced input embedding $\widetilde{x}^*$ will be fed into the foundation model backbone, thereby enhancing prediction accuracy.

## 3.5 KNOWLEDGE RETRIEVAL

Inspired by DPR (Karpukhin et al., 2020), we employ a dual-encoder retriever to efficiently obtain relevant information from the external knowledge base.

### 3.5.1 KNOWLEDGE RETRIEVAL LEARNING

The retriever adopts a MLP-based encoder to respectively embed the query and the candidates. In TimeRAF, we utilize the input directly as the query. Then, the retriever calculates the dot product similarity score between the query and each candidate using their respective embeddings. Finally, the candidates with the $k$ highest similarity scores are retrieved, denoted as $C = \{c_1, c_2, \ldots, c_k\}$.

Intuitively, by augmenting the model with retrieved knowledge, the goal is to improve predictions based on desired metrics, such as Mean Squared Error (MSE). However, it is challenging to guarantee that retrieved candidates with higher similarity scores will consistently provide more useful knowledge for forecasting. To address this, we employ the foundation model $\mathcal{F}$ as an evaluator, leveraging its strong forecasting capability to provide feedback and guide the selection of knowledge.

Specifically, using the retrieved candidate $c_i$, we employ the foundation model $\mathcal{F}$ to obtain the the metric values of prediction $\hat{y} = \mathcal{F}(x, c_i)$. If $\mathcal{F}$ finds that integrating the knowledge from $c_i$ is beneficial for forecasting, we encourage the retriever to rank the score of $c_i$ to be higher. In this way, the model can automatically decide the usefulness of the candidates and learn to retrieve more helpful candidates from the knowledge base. To implement this learning strategy, we first transform the metric values into a probability distribution as:

$$p_i = \frac{\exp(\frac{1}{\tau_m} \text{M}(\mathcal{F}(x, c_i), y))}{\sum_{j=1}^{k} \exp(\frac{1}{\tau_m} \text{M}(\mathcal{F}(x, c_j), y))}, \tag{3}$$

where $\text{M}(\hat{y}, y)$ denotes the metric function to evaluate the quality of the prediction $\hat{y}$ given the ground truth $y$ and $\tau_m$ is a temperature hyperparameter to control the sensitivity of the metric. Here the metric function satisfies that a higher value of $\text{M}(\cdot, \cdot)$ indicates better performance. If a smaller value of $\text{M}(\cdot, \cdot)$ indicates better performance, we can replace $\text{M}(\cdot, \cdot)$ with $-\text{M}(\cdot, \cdot)$ in equation 3.

It is evident that a beneficial $c_i$ will correspond to a higher $p_i$, allowing $p_i$ to serve as a supervised signal to guide the learning of the retriever. In particular, we aim to align the similarity score generated by the retriever with $P = \{p_i\}_{i=1}^{k}$. Formally, suppose we have top-k retrieval candidates $C_q$ along with their associated retrieval scores $S_q \in \mathbb{R}^k$ with respect to the query $q$. We can then aim to minimize the Kullback-Leibler divergence between $S_q$ and $P$ as follows:

$$\mathcal{L}_R = D_{\text{KL}}(P, \text{softmax}(S_q / \tau_s)), \tag{4}$$

where $D_{\text{KL}}$ denotes the KL divergence and $\tau_s$ is a temperature hyperparameter to control the sensitivity of the similarity scores.

| | Statistics | Energy | Transport | Nature | Web | Sales | Healthcare |
|---|---|---|---|---|---|---|---|
| Dataset | # Datasets | 3 | 2 | 2 | 2 | 2 | 2 |
| | # Obs | 10,875,374 | 8,223,748 | 9,784,137 | 157,104,689 | 58,411,778 | 72,583,275 |
| | % | 3.43% | 2.59% | 3.09% | 49.56% | 18.43% | 22.90% |
| Knowledge Base | # Datasets | 3 | 2 | 2 | 2 | 2 | 2 |
| | # Obs | 545,792 | 585,216 | 513,024 | 512,000 | 484,864 | 512,512 |
| | % | 17.31% | 18.56% | 16.27% | 16.24% | 15.38% | 16.25% |

Table 1: Key statistics of dataset and knowledge base by domain. # Datasets refers to the number of datasets and # Obs denotes the number of observable data points.

However, during the training process, there is a risk that the retriever may become entrenched in a local optimum, thereby consistently retrieving a limited set or a narrow range of candidates. Consequently, the forecaster fails to learn from the retriever and disregards the retrieved knowledge. To address this issue, we employ a straightforward augmentation strategy by incorporating randomly sampled data from the knowledge base to promote a broader exploration of candidates within the framework. Specifically, we initially replace each $c_i$ with a randomly selected candidate $c_i^{\mathrm{aug}}$ at a probability of $\rho$, yielding $C_q^{\mathrm{aug}}$. Then the dot product similarity between the query $q$ and each candidate $c_i^{\mathrm{aug}}$ will be updated as the retrieval scores $S_q^{\mathrm{aug}} = \{s_i^{\mathrm{aug}}\}_{i=1}^k$. Finally, based on equation 4, we can minimize the following loss to update the retriever:

$$\mathcal{L}_R^{\mathrm{aug}} = D_{\mathrm{KL}}(\boldsymbol{P}^{\mathrm{aug}}, \mathrm{softmax}(\boldsymbol{S}_q^{\mathrm{aug}}/\tau_s)). \tag{5}$$

### 3.5.2 RETRIEVER-FORECASTER JOINT TRAINING

Utilizing the candidates retrieved by the retriever, we aim to enhance forecasting capability by leveraging external knowledge and further supervising the training of the forecaster. As illustrated in Figure 2, the backbone of foundation model remains frozen throughout the training process. To maintain consistency, we employ the same prediction loss utilized during the pre-training phase to update the entire forecaster. Formally, the prediction loss can be formulated as follows:

$$\mathcal{L}_{\mathrm{Pred}} = \mathcal{L}_{\mathrm{Pretrain}}(\mathcal{F}(\boldsymbol{x}, \boldsymbol{C}), \boldsymbol{y}). \tag{6}$$

Combined with the loss utilized for updating the retriever, the whole training loss is

$$\mathcal{L} = \mathcal{L}_{\mathrm{Pred}} + \lambda \cdot \mathcal{L}_R^{\mathrm{aug}}, \tag{7}$$

where $\lambda$ is a weight hyperparameter of $\mathcal{L}_R^{\mathrm{aug}}$.

### 3.6 INFERENCE PROCEDURE

During the inference process, given a query, candidates with the highest $k$ retrieval scores from the knowledge base are retrieved by the retriever. Following preprocessing, the embeddings of the input and the retrieved candidates are processed through Channel Prompting to effectively integrate external knowledge. Ultimately, the knowledge-enhanced embeddings are fed into the backbone of the time series foundation model, which then generates the final prediction.

## 4 EXPERIMENT

### 4.1 EXPERIMENTS SETUPS

**Datasets and Knowledge Base:** Our training employs a subset of about 320 million time points from LOTSA (Woo et al., 2024) and UTSD (Liu et al., 2024), which were used for the pre-training of Time Series Foundation Models. The dataset encompasses a diverse range of domains to maintain the generalization capabilities of the foundation model. The knowledge base used for training contains approximately 3 million data points, as introduced in section 3.3, selected from the training datasets. Each domain within the knowledge base is designed to contain a roughly equivalent number of data points to maintain balance. Detailed statistics of our training dataset and knowledge base are provided in Table 1. Consistent with LOTSA, we adopt Arrow (Richardson et al., 2024) as the unified storage

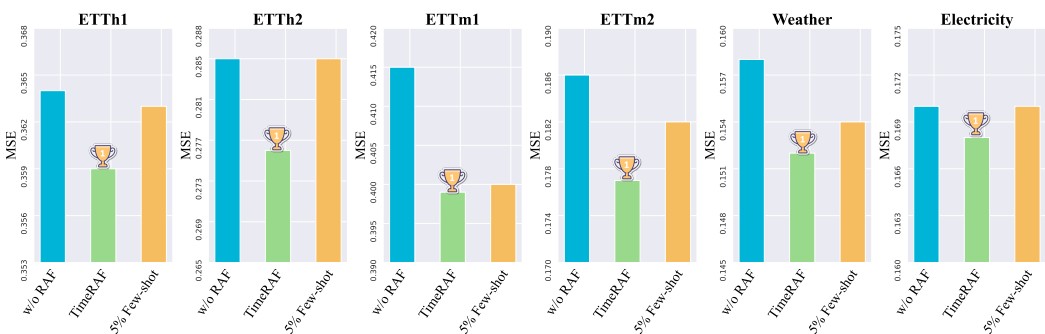

Figure 3: Improvement by TimeRAF on zero-shot forecasting. 5% Few shot denotes finetuning TSFM with 5% of downstream dataset. TimeRAF demonstrates significant improvements across various datasets, even outperforming results obtained by few-shot fine-tuning.

format, which is suitable for deep learning pipelines. For evaluation, we consider the popular long sequence forecasting benchmark, including six public datasets : ETTh1, ETTh2, ETTm1, ETTm2, Weather, and Electricity, which are commonly utilized in previous works (Jin et al., 2023; Woo et al., 2024). It should be noted that all evaluation datasets are inaccessible during the training process. The detail of our training datasets and evaluation datasets are provided in Appendix A.

**Metric:** We employ mean squared error (MSE) as the standard error metric for our experiments.

**Implementation Detail:** We employ TTM-Base (TTM$_B$), one of the latest State of The Art (SOTA) TSFM, as our backbone. The input context length is set to 512 and the forecasting length is 96, consistent with TTM$_B$. During inference, TimeRAF uses the same knowledge base employed during the training phase. More implementation details are provided in Appendix B.

**Baselines:** We compare with 12 of the latest open-sourced state-of-the-art forecasting methods categorized as follows: (a) Time Series Foundation Model: TTM (Ekambaram et al., 2024), Moirai (Woo et al., 2024), MOMENT (Goswami et al., 2024), Timer (Liu et al., 2024), Chronos (Ansari et al., 2024), TimesFM (Das et al., 2024). (b) LLM-based Time Series Model: TimeLLM (Jin et al., 2023), GPT4TS (Zhou et al., 2023). (c) Other architectures: iTransformer (Liu et al.), TimesNet (Wu et al., 2023a), PatchTST (Nie et al., 2023) and DLinear (Zeng et al., 2023). All results are sourced from Liu et al. (2024) or our reproduction. Due to page limitation, we report the results of the base version of TSFMs in the main text. Full results are provided in Appendix C.1.

## 4.2 RESULTS OF ZERO-SHOT FORECASTING

**Improvement by TimeRAF on zero-shot forecasting:** As shown in Figure 3, we demonstrate the improvements brought by our method in zero-shot forecasting. The yellow bar represents the scenario where 5% of the training data from the dataset is used to fine-tune the foundation model backbone. Augmented by retrieved knowledge, our TimeRAF presents significant improvements across all the datasets. The experiment results indicate that, through our training, our retriever has learned to search for valuable information from the knowledge base. Subsequently, through channel prompting, TimeRAF successfully extracts useful knowledge, ultimately enhancing the prediction results. Moreover, TimeRAF also outperforms the performance achieved through few-shot fine-tuning, which further demonstrate the effectiveness of our method.

**TimeRAF vs. other models:** We compare TimeRAF against 12 baseline models. The experiment results are shown in Table 2, where 'zero-shot' refers to the forecasting results of various foundation models without any prior training on the test datasets, while 'full-shot' represents the prediction results of baseline models that have been fully trained on each dataset. Compared to the foundation models, TimeRAF achieves either the best or competitive results across multiple datasets. Besides, our method is an enhancement built upon the foundation model. As the foundation model continues to evolve, TimeRAF is anticipated to yield further improvements when adapted to new backbones. Additionally, we observe that TimeRAF achieves strong results compared to full-shot baselines, thereby underscoring the effectiveness of retrieval-augmented forecasting.

Table 2: Full results of long sequence forecasting experiments. Best results are highlighted in **bold** and second best results are underlined. Besides, best results of zero-shot marked in Red.

| Dataset | Zero-shot | | | | | | | Full-shot | | | | | |
|---------|-----------|------|---------|--------|---------|--------|-----------|---------|-------|-------------|---------|---------|---------|
| | TimeRAF | TTM$_B$ | Moirai$_B$ | MOMENT | Timer$_{1B}$ | TimesFM | Chronos$_{S1}$ | TimeLLM | GPT4TS | iTransformer | TimesNet | PatchTST | DLinear |
| ETTh1 | 0.359 | 0.364 | 0.383 | 0.674 | 0.438 | 0.414 | 0.571 | 0.362 | 0.376 | 0.386 | 0.384 | 0.414 | 0.386 |
| ETTh2 | 0.276 | 0.285 | 0.295 | 0.330 | 0.314 | 0.318 | 0.423 | **0.268** | 0.285 | 0.297 | 0.340 | 0.302 | 0.333 |
| ETTm1 | 0.399 | 0.415 | 0.448 | 0.670 | 0.690 | 0.354 | 0.632 | **0.272** | 0.292 | 0.334 | 0.338 | 0.329 | 0.345 |
| ETTm2 | 0.177 | 0.186 | 0.225 | 0.257 | 0.213 | 0.201 | 0.272 | **0.161** | 0.173 | 0.180 | 0.187 | 0.175 | 0.193 |
| Weather | 0.152 | 0.158 | 0.197 | 0.255 | 0.181 | - | - | **0.147** | 0.162 | 0.174 | 0.172 | 0.177 | 0.196 |
| Electricity | 0.168 | 0.170 | 0.162 | 0.744 | 0.192 | - | - | **0.131** | 0.139 | 0.148 | 0.168 | 0.195 | 0.197 |

Table 3: Ablation Studies on Retriever and Channel Prompting. *Random* signifies the random selection while *Cosine* refers to retrieval based on cosine similarity. *Token-Concat* represents concatenating the retrieved candidates with the input at the token level. *Average* denotes directly computing the mean of the candidate and input embeddings. The best results are highlighted in **bold**.

| Dataset | TimeRAF | Random | Cosine | Token-Concat | Average |
|---------|---------|--------|--------|--------------|---------|
| ETTh1 | **0.359** | 0.365 | 0.360 | 0.363 | 0.367 |
| ETTh2 | **0.276** | 0.287 | 0.282 | 0.278 | 0.292 |
| ETTm1 | **0.399** | 0.420 | 0.401 | 0.404 | 0.421 |
| ETTm2 | **0.177** | 0.188 | 0.184 | 0.180 | 0.190 |
| Weather | **0.152** | 0.159 | 0.153 | 0.153 | 0.166 |
| Electricity | **0.168** | 0.173 | 0.172 | 0.174 | 0.181 |

## 4.3 ABLATION STUDIES

### 4.3.1 EFFECTIVENESS OF THE RETRIEVER

As described in section 3.5, we employ an end-to-end approach to train the retriever, encouraging it to select the most valuable candidates from the knowledge base. To validate the effectiveness of the learnable retriever, we have designed two baselines for comparison: one that randomly selects candidates from the knowledge base and another that selects the top $k$ candidates based on cosine similarity. As shown in Table 3, randomly selecting candidates fails to provide useful information to the forecaster and may even introduce noise, degrading the model's predictive performance. While the cosine similarity-based retrieval method offers some knowledge, its improvement is limited and falls short compared to our method, which automatically learns how to retrieve useful knowledge.

### 4.3.2 CHANNEL PROMPTING

An effective integration method, named Channel Prompting, is used to extract the relevant knowledge from the retrieved data, as detailed in section 3.4. To validate the effectiveness of channel prompting, we establish two baselines for comparison: the first, called *Token-Concat*, entails concatenating the retrieved candidates with the input at the token level, while the second, termed *Average*, involves directly computing the mean of the candidate and input embeddings for integration. As shown in Table 3, our TimeRAF outperforms both baselines. The token-level concatenation imposes restrictions on the integration to tokens located in the same position. While averaging input and retrieved candidates embeddings prove insufficient for extracting valuable information.

## 4.4 MODEL ANALYSIS

### 4.4.1 CHOICE OF KNOWLEDGE BASE

**Source of Knowledge Base:** The previous experimental results have convincingly demonstrated that following training, TimeRAF has acquired the capability to dynamically access pertinent knowledge from an external knowledge base and effectively leverage this valuable information. To further explore the implications of employing various knowledge bases during inference, we have devised the following three scenarios: (a) **TimeRAF$_R$** randomly selects data from the pre-trained multi-domain dataset, which may result in an uneven distribution across different domains. (b) **TimeRAF$_D$** utilizes a knowledge base closely related to the test data. Specifically, the training set from the same dataset is

Table 4: Comparison of TimeRAF with various knowledge bases. TimeRAF$_R$ employs random multi-domain data as the knowledge base and TimeRAF leverages the curated multi-domain knowledge base used for training. TimeRAF$_D$ utilizes a single-domain knowledge base, which differs from the one used during training but aligns with the domain of the test data. Best results are highlighted in **bold** and second best results are underlined.

| Dataset | w/o RAF | TimeRAF$_R$ | TimeRAF | TimeRAF$_D$ |
|---|---|---|---|---|
| ETTh1 | 0.364 | 0.362 | **0.359** | 0.360 |
| ETTh2 | 0.285 | 0.280 | **0.276** | 0.278 |
| ETTm1 | 0.415 | 0.404 | **0.399** | 0.400 |
| ETTm2 | 0.186 | 0.181 | **0.177** | 0.178 |
| Weather | 0.158 | 0.155 | **0.152** | **0.152** |
| Electricity | 0.170 | 0.169 | **0.168** | **0.168** |

directly employed as the knowledge base for retrieval. (c) **TimeRAF** engages a meticulously curated multi-domain dataset, which is detailed in Table 1. As presented in Table 4, TimeRAF achieves the best performance across different datasets. As a specifically designed knowledge base, it encompasses a rich repository of information across multiple domains, enabling it to provide useful information to enhance predictions. Meanwhile, the knowledge base used in TimeRAF$_D$ is particularly relevant to the test data, providing domain-specific knowledge. As a result, the zero-shot time series forecasting performance achieved with this knowledge base ranks just below that of TimeRAF. However, the randomly selected knowledge base used in TimeRAF$_R$ suffers from domain imbalance, which limits the enhancement that external knowledge can provide to the forecaster.

**Size of Knowledge Base:** The size of knowledge base also plays a vital role in the framework, determining the extent of external knowledge that can be accessed. We perform a comprehensive analysis of this aspect, presenting average results across various datasets in Figure 4. Full results are provided in Appendix C.3. Initially, both TimeRAF and TimeRAF$_D$ utilize knowledge bases of comparable scale, each consisting of approximately 3 million data points, as outlined in section 4.1. Then, we progressively reduce the size of the knowledge base. As shown in Figure 4, the MSE are influenced by modifications in the knowledge base size. As the size diminishes, the amount of external knowledge it can provide decreases, leading to a decline in the performance. Once the knowledge base is reduced beyond a certain point, using a domain-specific knowledge base (TimeRAF$_D$) can provide more relevant information compared to a multi-domain knowledge base (TimeRAF), resulting in better forecasting performance.

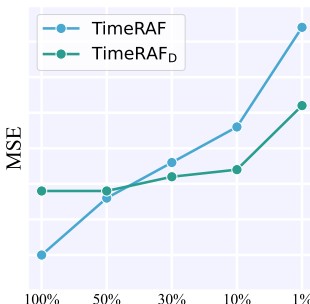

Figure 4: Influence of knowledge base size. Smaller knowledge base provides less information, leading to worse performance.

### 4.4.2 INFLUENCE OF THE CANDIDATES NUMBER

We investigate the impact of varying the numbers of retrieved candidates on prediction performance. As illustrated in Figure 5, using multiple retrieved candidates (e.g., 4 or 8) equips the forecaster with a more comprehensive set of external information compared to relying on a single candidate, thereby further enhancing prediction performance. Nevertheless, the performance gains do not persist as the variable $k$ increases. In our analysis of the test data, we observe that when $k$ is elevated to 16 or 32, there is no significant improvement in the model's prediction accuracy. This phenomenon may be attributable to the introduction of excessive candidates, which can lead to redundant information, ultimately detracting from the overall effectiveness of the prediction results.

### 4.5 CASE STUDY ON RETRIEVED KNOWLEDGE

To conduct a detailed analysis of the information provided by the retriever and its contribution to enhancing the zero-shot forecasting capabilities of the foundation model, we present two illustrative examples in Figure 6. As shown in Figure 6(a), the retrieved knowledge exhibits similar periodicity

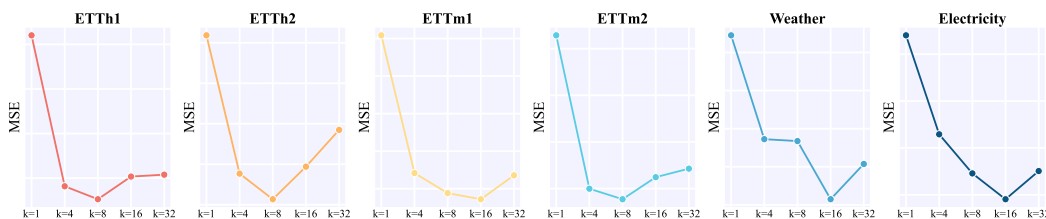

Figure 5: Influence of the Candidates Number $k$. As $k$ increases, the performance gradually improves due to the integration of more relevant knowledge. However, when $k$ exceeds a certain threshold, the abundance of information can introduce redundancy, negatively affecting the prediction.

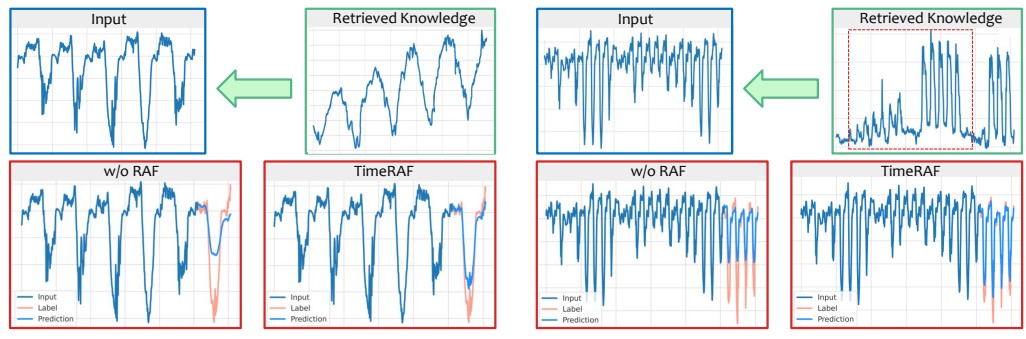

(a) Example A                                (b) Example B

Figure 6: **Case Study on Retrieved Knowledge. (a) Example A:** The retrieved knowledge shares similar periodicity and subtle fluctuations with the input, facilitating the forecaster's ability to effectively capture the prior knowledge inherent in the input, thereby improving prediction performance. **(b) Example B:** The retrieved data provides supplementary insights, including partial future information (highlighted within the red dashed box), empowering the forecaster to generate better predictions.

and nuanced fluctuations to the input, enhancing the forecaster's capacity to effectively capture the prior knowledge inherent in the input data, thereby improving prediction performance.

The data retrieved by the retriever is not always highly similar to the input, as illustrated in Figure 6(b). In the absence of the retrieval-augmented forecasting method, the model generates predictions with small amplitude, relying solely on constrained historical data and underlying inertia. However, the incorporation of retrieved data provides additional insights, including partial future information (highlighted within the red dashed box), thereby improving the prediction generated by the forecaster.

## 5 CONCLUSION AND FUTURE WORK

In this paper, we introduce TimeRAF, a novel framework designed to leverage retrieval-augmented generation for zero-shot time series forecasting. We develop customized time series knowledge bases that are tailored to the specific forecasting tasks and employ an end-to-end learnable retriever to extract valuable information from the knowledge base. We also introduce Channel Prompting to extract relevant information from the retrieved data for knowledge integration. By leveraging external knowledge, TimeRAF exhibits a notable enhancement in zero-shot time series forecasting.

While TimeRAF achieves phenomenal performance, this represents merely the initial step in the integration of time series methods and RAG. Due to resource constraints, the knowledge base is established based on original time series data data without the implementation of advanced techniques like trend-seasonal decomposition. In terms of architecture, our approach to integrate external knowledge is somewhat heuristic and future work should design a more flexible and elegant approach. Also, the current architecture has ignored the potential interdependencies among different channels, which could be addressed more effectively in future methods. Finally, incorporating multi-modality such as tabular or text data is an exciting new direction to provide supplementary knowledge.

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

# TimeRAF: Retrieval-Augmented Foundation model for Zero-shot Time Series Forecasting

# —————Appendix—————

CONTENTS

# A TimeRAF Datasets

## A.1 List of Training Datasets

Our fine-tuning employs a subset of about 320 million time points from LOTSA (Woo et al., 2024) and UTSD (Liu et al., 2024). To enhance data integrity, missing values are systematically addressed using linear interpolation techniques. For each univariate, multivariate, or irregular-sampled time series, we store them with timestamps, domains, sampling frequencies and other meta-information in one directory using ARROW format. One dataset may composed of multiple related time series.

All datasets can be classified into six distinct domains by their source: Energy, Nature, Transport, Web, Sales, and Healthcare. The datasets exhibit diverse sampling frequencies, ranging from macro intervals such as daily to more fine-grained intervals like hourly and minutely. Notably, several datasets can demonstrate exceptionally high-frequency sampling rates, such as the MotorImagery dataset, which operates at a millisecond frequency.

## A.2 List of Inference Datasets

In the field of time series forecasting, several classical datasets such as ETT (Zhou et al., 2021), ECL (Wu et al., 2021) and Weather (Wu et al., 2021) have become widely recognized benchmarks for evaluating model performance. We also utilize these datasets to evaluate the zero-shot forecasting performance and perform evaluation in a sliding window fashion following previous work (Nie et al., 2023; Ekambaram et al., 2024). Below, we offer a brief overview of these datasets.

1. **ETT datasets:** The four ETT datasets (ETTH1, ETTH2, ETTM1, ETTM2) contain multivariate time series data collected from electrical transformers at two stations. ETTH1 and ETTH2 are collected at an hourly interval, while ETTM1 and ETTM2 are collected every 15 minutes. All four datasets have 7 channels.

2. **Weather:** The weather dataset consists of 21 channels, which serve as weather indicators. It is collected at 10-minute intervals at the Max Planck Institute of Biogeochemistry weather station.

2. **Electricity (ECL):** The Electricity dataset, also known as the ECL dataset, comprises the hourly electricity consumption data of 321 clients.

# B Additional Implementation Detail

## B.1 Knowledge Base for Training

The knowledge base used for training contains approximately 3 million data points, as introduced in section 3.3, selected from the training datasets. The key statistics of datasets in knowledge base is provided in Table 1. The selection process entailed a meticulous curation to ensure a diverse representation of data across various domains. This diversity enhances the robustness of the model, enabling it to generalize better across different contexts. Each data point has been sourced from reputable datasets, ensuring high-quality input that informs the training process. Besides, all the time series data in the knowledge base has the same length as input, which is $512$.

## B.2 Training Details

Our training is performed on 4 NVIDIA A100 GPUs. Following the backbone configuration of the foundation model we use (TTM$_B$ (Ekambaram et al., 2024)), the input length $sl = 512$ and $fl = 96$. Both encoders in the retriever employ a two-layer MLP, with the size of the hidden layer configured to be four times the input dimension. The tanh activation function is utilized in both layers. The MLP in the Channel Prompting also use tanh activation function and has 4 layers. During the training phase, all parameters of the time series foundation model remain fixed, with only the retriever and channel prompting module undergoing training. The learning rate for the retriever is established at $0.001$, whereas the learning rate for the channel prompting is set at $0.00001$. The weight $\lambda$ is set to 1. The entire model was trained for 2 epochs, and for different test datasets, we reported the best results.

| Domain | Dataset | Frequency | Time Points | Source |
|---|---|---|---|---|
| Energy | BDG-2 Fox | H | 2,324,568 | BuildingsBench (Emami et al., 2023) |
| | Australian Electricity Demand | 30T | 1,153,584 | Monash (Godahewa et al., 2021) |
| | Solar Power | 4S | 7,397,222 | Monash (Godahewa et al., 2021) |
| Transport | Los-Loop | 5T | 7,094,304 | LibCity (Jiang et al., 2023) |
| | Uber TLC Hourly | H | 1,129,444 | GluonTS (Alexandrov et al., 2020) |
| Nature | Subseasonal Precipitation | D | 9,760,426 | SubseasonalClimateUSA library (Mouatadid et al., 2024) |
| | Saugeen | D | 23,711 | Monash (Godahewa et al., 2021) |
| Web | Kaggle Web Traffic Daily | D | 116,485,589 | Monash (Godahewa et al., 2021) |
| | Wiki-Rolling | D | 40,619,100 | GluonTS (Alexandrov et al., 2020) |
| Sales | M5 | D | 58,327,370 | GluonTS (Alexandrov et al., 2020) |
| | Favorita Transactions | D | 84,408 | Kaggle |
| Healthcare | MotorImagery | 0.001S | 72,576,000 | UCR Time Series Archive (Dau et al., 2019) |
| | US Births | D | 7,275 | Monash (Godahewa et al., 2021) |

Table 5: **Dataset detailed descriptions.** *Time Points* denotes the total number of time points aggregating from all variates if multivariate. *Frequency* denotes the sampling interval of time points. *Source* denotes the original paper or resource of the dataset.

During the inference phase, with the exception of the ablation experiments detailed in section 4.4.2, we consistently retrieved eight candidates from the knowledge base, where $k = 8$.

## B.3 BASELINES

We conduct zero-shot forecasting experiments on seven datasets from iTransformer (Liu et al.). We apply the same data-split strategy as Autoformer (Wu et al., 2021) and calculate the averaged MSE of all predict-96 windows in the test split. We evaluate five open-source time series foundation model, including Timer (Liu et al., 2024), Moirai (Woo et al., 2024), TimesFM (Das et al., 2024), Chronos (Ansari et al., 2024), and MOMENT (Goswami et al., 2024). However, closed-source models such as TimeGPT (Garza & Mergenthaler-Canseco, 2023) are not included due to their inaccessibility.

- **MOMENT**: MOMENT[1] employs a masking modeling approach for zero-shot forecasting by concatenating the lookback series with a mask corresponding to the prediction length. The output of the model, derived from the mask, serves as the forecast. This method involves pre-training a Transformer encoder model in a univariate manner using a curated dataset known as the "Time Series Pile," which encompasses a diverse range of time series data.
- **Chronos**: Chronos[2] is a probabilistic forecaster. *Chronos_{S1}* refers to sampling a single prediction trajectory, while *Chronos_{S20}* involves averaging 20 sampled trajectories. Chronos tokenizes the input time series and processes these tokens using a large language model, specifically the T5 model. It is trained on an extensive corpus of time series data, including synthetic data, to enhance generalization.
- **TimesFM**: TimesFM employs a decoder-style attention model, characterized by causal self-attention, which is pre-trained in a univariate manner on an extensive array of both real-world and synthetic datasets. We utilize the official checkpoint available on HuggingFace[3], which accommodates a variety of input and output lengths.
- **Moirai**: The Moirai family[4] has three different sizes, labeled as *Moirai_S*, *Moirai_M*, and *Moirai_L*. Moirai pre-trains a Transformer encoder on the extensive "LOTSA" dataset (27B time points) by masking the forecast horizon of each target channel and performing mask reconstruction. By flattening all channels in a multivariate time series, Moirai supports pre-training in "any-variate" settings.

[1] https://huggingface.co/AutonLab/MOMENT-1-large
[2] https://huggingface.co/amazon/chronos-t5-large
[3] https://huggingface.co/google/timesfm-1.0-200m
[4] https://huggingface.co/collections/Salesforce/moirai-10-r-models-65c8d3a94c51428c300e0742

- **Timer**: Timer provides three versions with increased scopes of pre-training. $Timer_{1B}$ is pre-trained on UTSD[5]; $Timer_{16B}$ is pre-trained on UTSD and Buildings900K (Emami et al., 2023); and $Timer_{28B}$ is pre-trained on UTSD and LOTSA.
- **TTM**: TTM[6] pre-trains a compact model based on the light-weight TSMixer architecture, incorporates innovations like adaptive patching, diverse resolution sampling, and resolution prefix tuning on Monash and LibCity datasets.

We report the implementation details for all the time series foundation model baselines in Table 6.

Table 6: Implementation details for time series foundation model baselines

| Baseline | Used in Table | Results Source | Implementation Link |
|---|---|---|---|
| Moirai | Zero-shot in Table 2 and Table 8 | Liu et al. (2024) | uni2ts |
| Timer | Zero-shot in Table 2 and Table 8 | Liu et al. (2024) | Large-Time-Series-Model |
| MOMENT | Zero-shot in Table 2 and Table 8 | Liu et al. (2024) | moment |
| Chronos | Zero-shot in Table 2 and Table 8 | Liu et al. (2024) | chronos-forecasting |
| TimesFM | Zero-shot in Table 2 and Table 8 | Liu et al. (2024) | TimesFM |
| TimesFM | Zero-shot in Table 2 and Table 8 | Our reproduction using official implementation | granite-tsfm |

Table 7: Quality evaluation of time series foundation models. *Architecture* denotes the Transformer category. *Model size* presents the parameter counts. *Token type* presents the graininess of time series tokens. *Context length* means the maximum/fixed input length of the model.

| Method | Timer (2024) | Moirai (2024) | MOMENT (2024) | Chronos (2024) | TTM (2024) | TimesFM (2024) |
|---|---|---|---|---|---|---|
| Model size | 29M, 50M, 67M | 14M, 91M, 311M | 40M, 125M 385M | 20M, 46M, 200M, 710M | 1M, 4M 8M | 17M, 70M, 200M |
| Supported tasks | Forecast Imputation Detection | Forecast | Forecast Imputation Classification Detection | Forecast | Forecast | Forecast |
| Pre-training Scale | 28B | 27.65B | 1.13B | 84B | 1B | 100B |
| Token type | Segment | Segment | Segment | Point | Segment | Segment |
| Context length | $\leq 1440$ | $\leq 5000$ | $= 512$ | $\leq 512$ | $\leq 1536$ | $\leq 512$ |
| Variable length | True | True | False | True | True | True |
| Probabilistic | False | True | False | True | False | True |

# C  ADDITIONAL EXPERIMENTS

## C.1  ZERO-SHOT FORECASTING EVALUATION

We provide zero-shot time series forecasting results of TimeRAF and other time series foundation model in Table 8. The results highlight the performance of TimeRAF in comparison to other leading time series foundation models, demonstrating its effectiveness in integrating external knowledge. This capability is particularly crucial for industries that require timely and reliable forecasting without the luxury of extensive historical data. Overall, the findings suggest that TimeRAF not only sets a new benchmark in zero-shot time series forecasting but also paves the way for future research on enhancing model architectures and training methodologies in this domain.

## C.2  ABLATION STUDY ON CANDIDATE AUGMENTATION

During the training process, there is a risk that the retriever may become entrenched in a local optimum, thereby consistently retrieving a limited set or a narrow range of candidates. To address

---

[5]https://huggingface.co/datasets/thuml/UTSD/tree/main

[6]https://huggingface.co/ibm-granite/granite-timeseries-ttm-v1

Table 8: Full results of zero-shot forecasting experiments. Best results are highlighted in **bold** and second best results are underlined.

| Dataset | TimeRAF | $TTM_B$ | $Moirai_S$ | $Moirai_B$ | $Moirai_L$ | MOMENT | $Timer_{1B}$ | $Timer_{16B}$ | $Timer_{28B}$ | TimesFM | $Chronos_{S1}$ | $Chronos_{S20}$ |
|---|---|---|---|---|---|---|---|---|---|---|---|---|
| ETTh1 | **0.359** | 0.364 | 0.441 | 0.383 | 0.394 | 0.674 | 0.438 | 0.364 | 0.393 | 0.414 | 0.571 | 0.34 |
| ETTh2 | **0.276** | 0.285 | 0.295 | 0.295 | 0.293 | 0.330 | 0.314 | 0.294 | 0.308 | 0.318 | 0.423 | 0.326 |
| ETTm1 | 0.399 | 0.415 | 0.562 | 0.448 | 0.452 | 0.670 | 0.690 | 0.766 | 0.420 | **0.354** | 0.632 | 0.451 |
| ETTm2 | **0.177** | 0.186 | 0.218 | 0.225 | 0.214 | 0.257 | 0.213 | 0.234 | 0.247 | 0.201 | 0.272 | 0.190 |
| Weather | **0.152** | 0.158 | 0.195 | 0.197 | 0.221 | 0.255 | 0.181 | 0.203 | 0.243 | - | - | - |
| Electricity | 0.168 | 0.170 | 0.212 | 0.162 | 0.155 | 0.744 | 0.192 | **0.139** | 0.147 | - | - | - |

Table 9: Ablation study on candidate augmentation

| Dataset | TimeRAF | TimeRAF w/o candidate augmentation |
|---|---|---|
| ETTh1 | **0.359** | 0.363 |
| ETTh2 | **0.276** | 0.282 |
| ETTm1 | **0.399** | 0.410 |
| ETTm2 | **0.177** | 0.186 |
| Weather | **0.152** | 0.161 |
| Electricity | **0.168** | 0.173 |

this issue, we employ a straightforward augmentation strategy. We provide experiments results of TimeRAF without candidate augmentation in Table 9.

## C.3 SIZE OF KNOWLEDGE BASE

Table 10: Influence of different knowledge base size. Best results are highlighted in **bold**.

| Dataset | TimeRAF | | | | | $TimeRAF_D$ | | | | |
|---|---|---|---|---|---|---|---|---|---|---|
| | 100% | 50% | 30% | 10% | 1% | 100% | 50% | 30% | 10% | 1% |
| ETTh1 | **0.3592** | 0.3598 | 0.3603 | 0.3608 | 0.3622 | 0.3599 | 0.3600 | 0.3601 | 0.3602 | 0.3611 |
| ETTh2 | **0.2763** | 0.2767 | 0.2773 | 0.2790 | 0.2844 | 0.2779 | 0.2785 | 0.2791 | 0.2804 | 0.2823 |
| ETTm1 | **0.3991** | 0.3995 | 0.4002 | 0.4017 | 0.4038 | 0.3998 | 0.4002 | 0.4008 | 0.4015 | 0.4024 |
| ETTm2 | **0.1768** | 0.1773 | 0.1778 | 0.1792 | 0.1815 | 0.1776 | 0.1780 | 0.1784 | 0.1791 | 0.1807 |
| Weather | **0.1522** | 0.1527 | 0.1533 | 0.1542 | 0.1558 | 0.1524 | 0.1528 | 0.1533 | 0.1540 | 0.1551 |
| Electricity | **0.1681** | 0.1686 | 0.1692 | 0.1701 | 0.1715 | 0.1684 | 0.1688 | 0.1691 | 0.1698 | 0.1710 |

We conduct an experiment to examine the impact of knowledge base size on performance. Initially, TimeRAF and $TimeRAF_D$ uses knowledge bases of identical scale, each comprising approximately 3 million data points, as detailed in section 4.1. We progressively reduce the size of the knowledge base and valuate the task. As shown in Figure 1, the zero-shot forecasting results on the ETTh1 dataset vary with changes in the knowledge base size. As shown in the figure, when the knowledge base becomes smaller, the amount of external knowledge it can provide decreases, leading to a decline in prediction performance. Once the knowledge base is reduced beyond a certain point, using a domain-specific knowledge base can provide more relevant information compared to a multi-domain knowledge base, resulting in better forecasting performance.

## C.4 FORECASTING VISUALIZATION

We provide several visualization of zero-shot forecasting in Figure 7. These visualizations illustrate the effectiveness of our proposed method based on leveraging external knowledge. Each subplot in Figure 7 captures distinct scenarios, allowing for a comprehensive understanding of the model's capabilities under different conditions.

## C.5 MODEL EFFICIENCY COMPARISON

The comparison of model efficiency is presented in Table 11. The supplementary modules incorporated in RAF are designed to be lightweight, contributing only a minimal increase in parameters.

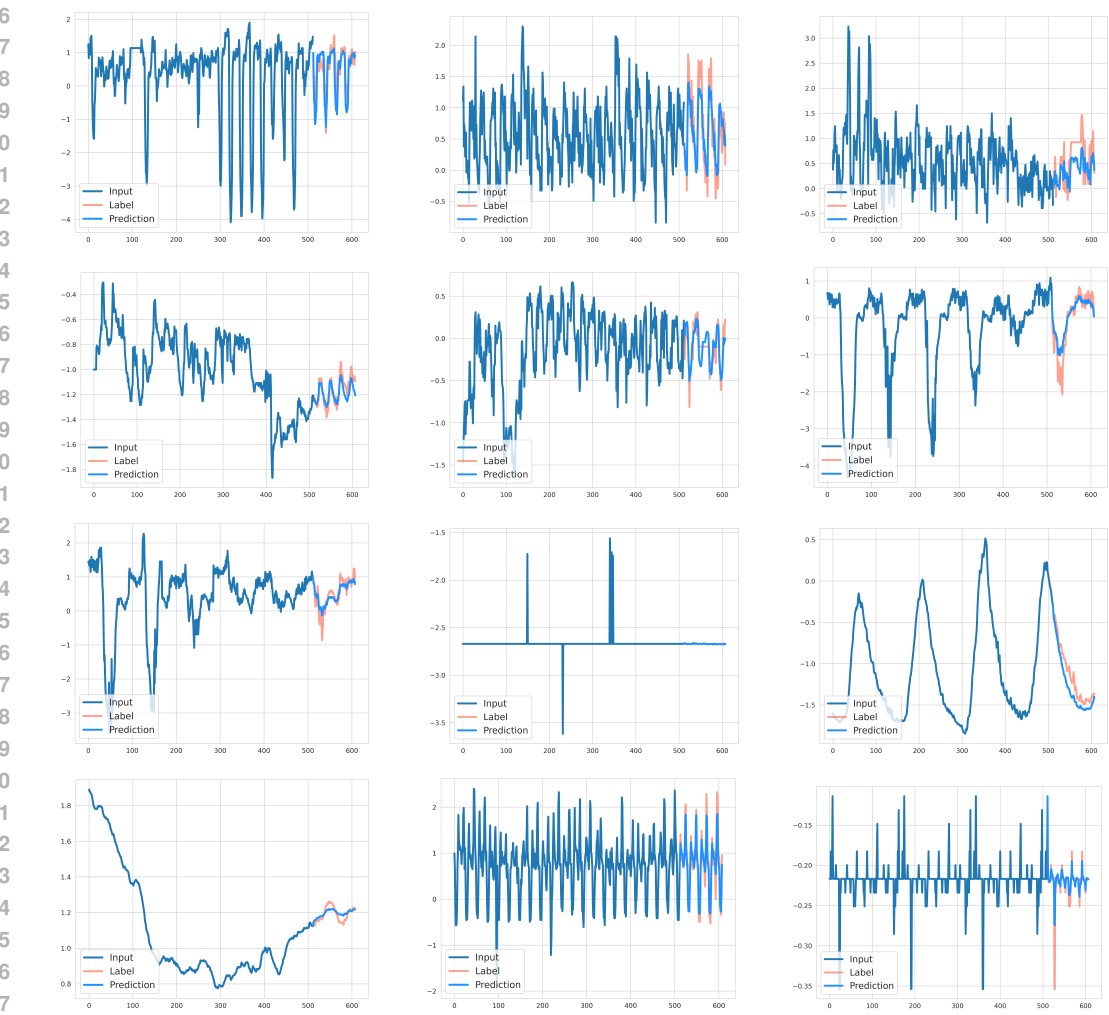

Figure 7: Visualization of zero-shot forecasting across different datasets.

|  | TimeRAF | TTM$_B$ | Moirai$_B$ | MOMENT | TimesFM | Chronos$_{S1}$ |
|---|---|---|---|---|---|---|
| Model Size | 8M | 1M | 91M | 348M | 200M | 8M |
| Inference Time | 0.13s | 0.01s | 3.7s | 1.4s | 0.4s | 2500s |

Table 11: **Model Efficiency Comparison:** We provide model size and per batch CPU inference time of each foundation model.

Furthermore, the retrieval process is based on a dot product calculation, which enhances efficiency. Consequently, TimeRAF maintains a satisfactory model size and inference time.

### C.6 MULTIPLE FORECAST HORIZONS

The full results of multiple forecast horizons are presented in Table 11. From these results, we observe that TimeRAF consistently delivers significant improvements over the backbone across all forecast horizons, further demonstrating the method's robustness and generality.

### C.7 ALTERNATIVE BACKBONE

| | | TimeRAF | w/o RAF |
|---|---|---|---|
| ETTh1 | 96 | **0.359** | 0.364 |
| | 192 | **0.384** | 0.387 |
| | 336 | **0.397** | 0.403 |
| | 720 | **0.457** | 0.475 |
| ETTh2 | 96 | **0.276** | 0.285 |
| | 192 | **0.335** | 0.346 |
| | 336 | **0.364** | 0.385 |
| | 720 | **0.408** | 0.419 |
| ETTm1 | 96 | **0.399** | 0.415 |
| | 192 | **0.377** | 0.380 |
| | 336 | **0.397** | 0.402 |
| | 720 | **0.442** | 0.446 |
| ETTm2 | 96 | **0.177** | 0.186 |
| | 192 | **0.238** | 0.246 |
| | 336 | **0.290** | 0.323 |
| | 720 | **0.386** | 0.406 |
| Weather | 96 | **0.152** | 0.158 |
| | 192 | **0.192** | 0.195 |
| | 336 | **0.251** | 0.256 |
| | 720 | **0.321** | 0.328 |
| Electricity | 96 | **0.168** | 0.170 |
| | 192 | **0.194** | 0.197 |
| | 336 | **0.212** | 0.214 |
| | 720 | **0.261** | 0.264 |

Table 12: **Full Forecasting Results:** TimeRAF consistently delivers significant improvements over the backbone across all forecast horizons

| | Timer w/o RAF | Timer w/ RAF | Improvements (%) |
|---|---|---|---|
| ETTh1 | 0.427 | 0.438 | 2.51 |
| ETTh2 | 0.305 | 0.314 | 2.87 |
| ETTm1 | 0.671 | 0.69 | 2.75 |
| ETTm2 | 0.203 | 0.213 | 4.69 |
| Weather | 0.173 | 0.181 | 4.42 |
| Electricity | 0.187 | 0.192 | 2.60 |

Table 13: **Zero-shot Forecasting Results:** Using Timer (Liu et al., 2024) as the backbone, TimeRAF consistently delivers significant improvements over the backbone across all datasets.

To further evaluate the effectiveness of RAF, we conducted additional experiments using Timer (Liu et al., 2024) as the backbone. The results are provided in the Table 13. These experiments demonstrate that our method consistently delivers significant improvements across different backbone models.

# D  ADDITIONAL DISCUSSION

## D.1  DISCUSSION ON CHANNEL PROMPTING

For each retrieved candidate $c_i$, we will extract the valuable feature from the combined embedding $z_i$ first. Then, to retain useful information from all $k$ retrieved sequences, the features extracted from these sequences are averaged. This ensures that the model captures information from all retrieved data while balancing their contributions. These two operations are complementary and work together to integrate the retrieved information effectively.

