# OpenReview forum: "TimeRAF: Retrieval-Augmented Foundation model for Zero-shot Time Series Forecasting"
_ICLR.cc/2025/Conference — ICLR 2025 Conference Withdrawn Submission_

### Official Review · Reviewer_U1MA · 2024-10-28

**Soundness:** 2
**Presentation:** 2
**Contribution:** 2
**Rating:** 5
**Confidence:** 4

**Summary:**

This paper introduces **TimeRAF**, a Retrieval-Augmented Forecasting model designed to enhance so-called zero-shot time series forecasting by leveraging retrieval-augmented techniques. The model integrates external time series knowledge bases to supplement forecasting tasks, employing an end-to-end learnable retriever.

**Strengths:**

1. The article is well-structured.
2. Some experiments are meaningful.

**Weaknesses:**

1. **Channel Prompting Efficiency Concern**:
   If I understand correctly, **Channel Prompting** requires each channel to process and fuse information from **k retrieved time series**. This may introduce significant computational overhead, especially for datasets with many channels (e.g., **traffic data**). The paper lacks an analysis of the efficiency and scalability of this mechanism in such scenarios.

2. **Lack of Pre-training Details for TSFM**:
   The paper omits essential information about the **pre-training process** of the time series foundation model (TSFM), such as the **datasets used, training procedure, and loss functions**. This raises concerns about the reproducibility and transparency of the method.

3. **Misinterpretation of Zero-Shot Learning**:
   I disagree with the zero-shot setup in the paper. True zero-shot learning implies that the model is tested on datasets it has **never seen during training**. For example, training on **ETTh1** and testing on **ETTh2** would align with this standard. However, as depicted in **Figure 2**, TimeRAF appears to fine-tune both the retriever and forecaster on parts of the test data, which **violates the principles of zero-shot forecasting**. The authors could refer to **GPT4TS or TimeLLM's zero-forecasting setup** for a more rigorous zero-shot methodology.

4. **Unfair Comparison in Experiments (Table 2)**:
   The experiments reported in **Table 2** are not entirely fair. According to **Section 4.1**, TimeRAF uses an input length of **512**. For fair comparison, **all baselines** should have the same input length. However, the reported metrics for baselines like **iTransformer, PatchTST, and DLinear** do not correspond to **input length = 512**. I have run these models with input lengths of **512**, and the results (e.g., PatchTST on **ETTm1** with 96-step forecast: 0.292, and **ETTm2**: 0.165) differ significantly from those reported in the paper.

5. **Incomplete Experimental Results**:
   The paper only reports results for a single forecast length (**96**). To comprehensively demonstrate the effectiveness of TimeRAF, it is essential to present results for **multiple forecast horizons** (e.g., **96, 192, 336, and 720**). This would provide a more complete evaluation of the model’s performance.

6. **Typos and Grammar Issues**:
   There are several typos and grammatical errors in the paper. For example:
     - **Lines 53-70**: “our exploration of Retrieval-Augmented for time series Forecasting (RAF)” is awkwardly phrased.
     - **Line 301**: “...updating the retriever,, the whole...” has an extra comma.
   The authors should carefully proofread the paper to improve its clarity and polish.

7. **Missing Axis Labels in Figure 5**:
   **Figure 5** lacks a **y-axis label**, which makes it difficult to interpret the results. Proper labeling is essential for readers to understand the figures accurately.

8. **Scaling Issue in Retrieval-Augmented Forecasting**:
   One important concern is how TimeRAF handles cases where the **retrieved time series data** are in **different magnitudes** than the input data. It would be useful if the authors could elaborate on any **normalization or scaling techniques** employed to address this issue.

9. **Implementation Details for Database Usage (Section 3.3)**:
   The paper mentions that **non-similar data** are used during training, while **similar data** are used during inference (Section 3.3, lines 187-193). However, the specific **implementation details** of how this is done for different datasets are unclear. Providing more insights here would enhance the paper's reproducibility and understanding.

10. **Lack of Code Availability**:
   The authors do not provide **code** for their experiments. Releasing code is crucial for **reproducibility** and would allow the community to verify the results and build upon the work.

**Questions:**

1. How scalable is the Channel Prompting mechanism for high-dimensional datasets with many channels?
2. How does TimeRAF ensure consistent scaling between retrieved external time series data and input data to avoid magnitude mismatches?
3. Why do the authors claim a zero-shot setup if the model fine-tunes on test data? How does this setup align with the standard definition of zero-shot forecasting?
4. What datasets and loss functions were used to pre-train the TSFM? How does the pre-training contribute to the model’s performance?
5. Can the authors provide results for other forecast lengths (e.g., 192, 336, and 720) to validate TimeRAF's generalization capability across multiple horizons?

---

> ### Author Response · Authors · 2024-11-19
> **Response by Authors (1/3)**
>
> Thank you for your feedback and the time invested in reviewing our work. We appreciate your insights and address your concerns as follows:
>
> 1. **Clarification on Channel Prompting:**
>
>     Thank you for raising this point. It seems there might have been a misunderstanding regarding our proposed channel prompting approach. We would like to take this opportunity to clarify and address any confusion.
>
>     Our entire framework is **channel-independent**—both the input and retrieved data are designed to operate with a single channel. This ensures that the approach does not introduce significant computational overhead, preserving the **efficiency** and **scalability** of our method.
>
>     To further clarify, the term *“channel”* in this context specifically refers to the channel of the **input embedding** and the **retrieved sequence’s corresponding embedding**, rather than different dimensions of multi-variant time series features.
>
>     We deeply regret any ambiguity in our initial explanation and have revised the manuscript to make this distinction clearer. Thank you for helping us identify this point of improvement—we truly appreciate your insights and constructive feedback.
>
> 2. **Pre-training Details for TSFM**:
>
>     Thank you for your feedback regarding the details of the pre-training process for the TSFM. We clarify that **we did not train a foundation model from scratch**; instead, we performed **fine-tuning** on an existing pre-trained model, such as TTM [1]. The specifics of the pre-trained model, including its **architecture** and **training setup**, are detailed in **Appendix B.3** of the original manuscript. Furthermore, the **fine-tuning** process and the adjustments we made are described in **Section 4.1 and Appendix B.2** of the original manuscript.
>
>     We understand the importance of transparency and reproducibility, which is why we have provided references and supplementary details. For more in-depth information about the original pre-training process of TTM, we kindly direct you to the original work [1]. We appreciate your understanding  and have strived to ensure the clarity of these aspects in our revised manuscript.
>
> 3. **Zero-shot Setup:**
>
>     Thank you for highlighting your concerns regarding our zero-shot learning setup. We greatly appreciate your attention to the rigor of experimental methodology. However, we believe there may have been a misunderstanding regarding our approach.
>
>     We fully understand and agree with your definition of zero-shot learning as testing on datasets that remain entirely unseen during training. **Our experimental setting adheres strictly to this principle: the test datasets are not used at any stage of training**, whether in pre-training or fine-tuning. In Appendix A.1, we present a comprehensive list of the datasets utilized for training, ensuring that there is no overlap with the test datasets. Specifically, we using the following datasets for training: BDG-2Fox,  Australian Electricity Demand,  Solar Power, Los-Loop, Uber TLC Hourly, Subseasonal Precipitation, Saugeen, Kaggle Web Traffic Daily, Wiki-Rolling, M5, Favorita Transactions, MotorImagery, and US Births, which are **totally different from the test datasets.**
>
>     Additionally, our zero-shot setup is **aligned with the standards established by recent works on time series foundation models**, **such as TTM [1] and Moirai [2]**. Specifically, the setup ensures that **test data is excluded in training datasets and unseen across a multi-dataset training phase**, **offering a more representative assessment of zero-shot capabilities for time series foundation models.**
>
>     We appreciate the opportunity to clarify this and will revise the manuscript to make our experimental setup clearer, ensuring no room for misinterpretation. Thank you for your thoughtful feedback and for pointing out ways we can improve the clarity of our work.

---

> ### Author Response · Authors · 2024-11-19
> **Response by Authors (2/3)**
>
> 4. **Comparison with baselines:**
>
>     Thank you for pointing out your concerns regarding the alignment of input lengths in our experiments. We greatly value your perspective and understand the importance of ensuring fairness in comparisons.
>
>     In our work, the results for baselines such as PatchTST, DLinear, and iTransformer **were directly referenced from well-established, peer-reviewed studies**, including Table 22 of Moirai [2] and Table 18 of Timer [3]. Our intention was to maintain consistency with prior benchmarks in the time series foundation model (TSFM) literature, where these results have been widely adopted. Additionally, the full-shot results were included in our work primarily to highlight the significant progress our method makes in **narrowing the gap between zero-shot and full-shot forecasting**.
>
>     We also note that current TSFM studies often use different input lengths across methods—such as 512 in TTM [1], 1000 in Moirai [2], and 672 in Timer [3]—yet these models are routinely compared in the zero-shot forecasting benchmark within the field. Therefore, we strictly adhered to the experimental settings of previous works to ensure a fair comparison.
>
>     Thank you again for your valuable feedback.
>
> 5. **Multiple forecast horizons**
>
>     Thank you for highlighting the importance of evaluating the model across multiple forecast horizons. This is indeed a valuable suggestion for providing a more comprehensive demonstration of TimeRAF's effectiveness. Due to the pretrained TTM model’s weight and code release constraints, **only the 96-step forecast horizon version is available prior to the submission deadline.** Consequently, we conducted our evaluations within this setup to ensure consistency and reliability in results. **With the recent release of additional versions for other forecast horizons (192, 336, and 720), we have conducted supplementary experiments.** We provide the average results below and have been added the full results to **Appendix C.6** of the revised manuscript:
>
>     |  | TimeRAF | w/o RAF | Improvement |
>     | --- | --- | --- | --- |
>     | ETTh1 | **0.399** | 0.407 | 1.96% |
>     | ETTh2 | **0.345** | 0.359 | 3.62% |
>     | ETTm1 | **0.404** | 0.411 | 1.71% |
>     | ETTm2 | **0.273** | 0.290 | 6.03% |
>     | Weather | **0.229** | 0.234 | 2.24% |
>     | Electricity | **0.209** | 0.211 | 1.18% |
>
>     From these results, we observe that TimeRAF consistently delivers significant improvements over the backbone across all forecast horizons, further demonstrating the method’s robustness and effectiveness.
>
>     We deeply appreciate your suggestion, as it allowed us to strengthen the empirical evidence supporting the proposed approach.
>
> 6. **Handling Scaling Differences in Retrieved Data:**
>
>     We fully agree with you that handling magnitude differences between retrieved and input time series data is crucial. **Actually, we observed this as well and addressed it by applying the standardization technique to the retrieved data, which is also applied to the input data**, to ensure alignment in magnitude. To clarify this point, we have updated the manuscript to specifically describe this scaling approach.

---

> ### Author Response · Authors · 2024-11-19
> **Response by Authors (3/3)**
>
> 7. **Implementation Details for Knowledge base:**
>
>     We apologize for any confusion regarding the description of data usage. To clarify, **our intention was not to imply retrieval of “non-similar data” or “similar data”.** Instead, we claimed that “**our retriever is designed to retrieve information solely from datasets that are distinct from the input source.”** Specifically, as described in Section 4.1, the knowledge base is built from multiple datasets within the training set, so **the retriever is restricted to draw from datasets other than the one associated with the input data during training**. For example, if both the training set and the knowledge base include data from **datasets A, B, C, and D**, and the input data comes from **dataset A**, our retriever will only search within **datasets B, C, and D**. This setup avoids the risk of retrieving future segments of the input data and ensures that the retrieval process during training does not lead to data leakage. **During inference**, since the test data has no overlap with training data or knowledge base, there is no additional constraints on the retrieval process. We have revised Section 3.3 to better convey this distinction and hope this addresses your concern.
>
> 8. **Code Availability：**
>
>     To facilitate reproducibility and allow the reviewer to examine our implementation in detail, we have provided an anonymous code [link](https://anonymous.4open.science/r/TimeRAF-77A3/README.md).
>
>
> Additionally, we thank the reviewer for pointing out the typos and grammatical issues and the labels of figures. We apologize for any inconvenience caused and have carefully proofread the manuscript to correct these errors.
>
> We hope these responses address your concerns and questions. Given that there may be some misunderstandings regarding our method details and experimental setup, which could have led to misconceptions, we kindly ask for your reconsideration of the score. Please feel free to reach out with any further questions, and we appreciate your thoughtful feedback, which has helped us strengthen the clarity and rigor of our work.
>
> References:
>
> [1] Ekambaram, Vijay, et al. "TTMs: Fast Multi-level Tiny Time Mixers for Improved Zero-shot and Few-shot Forecasting of Multivariate Time Series." *Advances in Neural Information Processing*, 2024.
>
> [2] Woo, Gerald, et al. "Unified Training of Universal Time Series Forecasting Transformers." *Forty-first International Conference on Machine Learning*.
>
> [3] Liu, Yong, et al. "Timer: Generative Pre-trained Transformers Are Large Time Series Models." *Forty-first International Conference on Machine Learning*.

---

> ### Author Response · Authors · 2024-11-27
> **Thank you for the review! Have we clearly addressed the concerns?**
>
> We greatly appreciate the time you took to review our paper. In the rebuttal, we have clarified the experimental setup and implementation detail of our method and provided more detailed explanation and additional experiments for our framework.
>
> Due to the short duration of the author-reviewer discussion phase, we would appreciate your feedback on whether your main concerns have been adequately addressed. We are ready and willing to provide further explanations and clarifications if necessary. Thank you very much!

---

> ### Author Response · Authors · 2024-11-30
>
> Dear Reviewer U1MA,
>
> Since the End of author/reviewer discussions is approaching, may we know if our response addresses your main concerns? If so, we kindly ask for your reconsideration of the score.
>
> Should you have any further advice on the paper and/or our rebuttal, please let us know and we will be more than happy to engage in more discussion and paper improvements. We would really appreciate it if our next round of communication could leave time for us to resolve any of your remaining or new questions.
>
> Thank you so much for devoting time to improving our methodology!

---

> > ### Comment · Reviewer_U1MA · 2024-11-30
> > **Sorry for remaining  serious concerns**
> >
> > Thank you for the author's reply. However, I am still filled with doubts about the experimental results and have strong reservations about the effectiveness of Timeraf.
> >
> > 1. To Response 1: The author explained the concept of channel prompting, but to avoid misunderstandings among readers, I suggest the author explain its working principle when introducing channel prompting. The explanation in the main text is indeed too brief, especially since the author treats it as a primary component of the method. Additionally, the author claims, “Our entire framework is channel-independent—both the input and retrieved data are designed to operate with a single channel. This ensures that the approach does not introduce significant computational overhead, preserving the efficiency and scalability of our method.” I believe the claim that channel-independence leads to efficiency and scalability is not valid. The result of channel-independence is that each multivariate time series requires querying channel * k external time series data, while without channel-independence, only k external time series are required (i.e., the same set of multivariate time series shares k external time series).
> >
> > 2. To Response 2: Thanks to your introduction, I now understand that the pretraining details of TSFM are primarily based on the TTM work and the pretrained model TTMB they provided. However, I am confused as to why the results reported by the author differ significantly from those in the original paper. The results of RAF+TTMB even seem worse than the TTMB results reported in the original TTM paper.
> >
> > 3. To Response 3: Thank you for the reply. I now understand that the zero-shot setup follows TTM.
> >
> > 4. To Response 4: The author's reply did not address my concern. The unfair comparison (due to different input lengths) greatly diminishes the reliability of the experimental results, as I have already explained in my initial inquiry.
> >
> > 5. To Response 5: The results provided by the author are not convincing. 1. The full scripts for all steps were not provided, so I cannot verify the experimental and ablation results. 2. The author’s backbone is TTMB, so the results without RAF should match those in the original TTM paper (e.g., Table 12), but they are far higher than the results reported in the original paper. What is the reason for this discrepancy? 3. The results reported with RAF do not show any improvement when compared to the TTMB results without RAF from the TTM paper, which makes me question the necessity and effectiveness of RAF.
> >
> > 6. To Response 6: In fact, the difference between the data is the main reason I believe that RAG cannot effectively integrate with time forecasting models. Even after normalization, there still exist significant differences in data distribution.
> >
> > 7. To Response 7: The author provided a detailed explanation that further improved the readability of the paper.
> >
> > 8. To Response 8: The author only provided the code and did not provide the complete running scripts to validate the ablation experiments and the authenticity of the results.
> >
> > In summary, because the results reported by the author are unfair, (1) many of the baseline comparisons do not use the same input length settings. (2) The author claims they used the TTMB backbone and pretrained parameters, but the results without RAF are far from the original TTMB version, which makes it difficult to trust. (3) The results reported with TTMB+RAF do not show improvement compared to the TTMB results in the original TTM paper and even seem worse.
> >
> > Therefore, despite the model description seeming reasonable, the untrustworthy experiments lead to doubt, and I cannot believe in the effectiveness of the author's model. Based on the above findings, i may decrease my scores to 1 to reflect on my concerns, but  it is too mean to do that and maybe you have some reasons, so i will stay for your next response.

---

> > > ### Author Response · Authors · 2024-12-01
> > > **Response to Further Concern**
> > >
> > > Thank you for your feedback and valuable suggestions! We appreciate your insights and address your further concerns as follows:
> > >
> > > 1. Efficiency about Channel Independence
> > >
> > >     Thank you for your insightful feedback and for pointing out this important consideration. Since our method is designed to be plug-and-play relative to the original backbone, we followed TTM's channel-independent strategy. Our intention was to emphasize that our approach does not introduce significant computational overhead when built on the existing backbone architecture. We apologize if our earlier statement caused any confusion.
> > >
> > >     In reference to the retrieval process without channel independence, as you mentioned, we agree that querying single-channel time series or fixed channels could result in fewer retrievals. However, this approach may introduce additional complexity when constructing the knowledge base, as it requires careful alignment between the number of channels in the knowledge base time series and the input data. Furthermore, combining the retrieved multivariate time series with the input multivariate time series in a meaningful way adds another layer of intricacy.
> > >
> > >     Your suggestion provides a valuable perspective on handling multivariate time series more effectively. As our work represents an initial exploration of integrating RAG with time series foundation models, we aimed to introduce a straightforward module based on a channel-independent backbone. We appreciate your constructive suggestion and believe that your proposed approach is indeed a promising direction for future research.
> > >
> > > 2. Comparison with full-shot methods
> > >
> > >     We appreciate your acknowledgment that our zero-shot setup follows TTM. Indeed, since we adhered to the TTM setup, as described in Appendix E.4 of the TTM paper, the results from the full-shot method in the Moirai paper are used for comparison.
> > >
> > >     Regarding your concern about the comparison's fairness due to differing input lengths: we understand and respect your point. However, we believe that a more critical comparison lies in the zero-shot results between different foundation models. Due to the inherent differences in the architectural designs of these models, it is challenging to unify the context lengths across all foundation models. As previously mentioned, models like Moirai, TTM, and Timer are often directly compared despite these variations.
> > >
> > >     We hope you can understand this limitation, and we acknowledge the need for a more comprehensive and fair benchmark tailored specifically for foundation models. Establishing such a benchmark is a valuable direction for future work, and we appreciate your feedback in highlighting this aspect. Thank you once again for your thoughtful comments and constructive suggestions!
> > >
> > > 3. Concern about the difference between data
> > >
> > >     Thank you for sharing your perspective. We agree that differences in data distribution persist even after normalization. However, we believe this does not preclude the effective integration of RAG with time series forecasting models, especially when leveraging foundation models trained on large-scale, diverse datasets. The scenario you described, where integration may fail, might be more applicable to models trained solely on a single data source.
> > >
> > >     Furthermore, to address the challenge of integrating RAG, we specifically designed a learnable retriever that utilizes feedback from prediction results to learn how to select data that is most beneficial for forecasting. We believe this design helps mitigate the issues associated with distribution differences to a certain extent. Additionally, Section 6 of [1] demonstrates that for LLMs, training on a mixed distribution of data sources leads to superior generalization on unseen RAG tasks compared to training on a single data source. Combined with our experimental results, we believe that integrating RAG with time series foundation models is both a feasible and promising direction for future exploration.
> > >
> > > We hope these responses address your concerns. We are ready and willing to provide further explanations and clarifications if necessary. Thank you very much!
> > >
> > > Reference:
> > >
> > > [1] Jin, Bowen, et al. "Long-Context LLMs Meet RAG: Overcoming Challenges for Long Inputs in RAG." *arXiv preprint arXiv:2410.05983* (2024).

---

> ### Comment · Reviewer_U1MA · 2024-12-01
> **Sorry for later reply, you can answer some critical question first.**
>
> For the author, if answering each of my questions individually would take too much time, you could first provide an explanation for the significant discrepancy between the Timeraf results and the TTMB results reported in the original TTM paper. I believe this is a critical issue.

---

> > ### Author Response · Authors · 2024-12-02
> > **Further Clarification and Compelling Evidence Supporting Our Reproduced Results**
> >
> > Dear Reviewer U1MA,
> >
> > We appreciate the reviewers' concerns regarding the differences between the results reported in the TTM paper and our reproduced experiments. We have conducted further investigations and uncovered compelling evidence supporting the reliability of our reproduced results.
> >
> > Specifically, we examined the official TTM repository and analyzed the provided [Jupyter notebook](https://github.com/ibm-granite/granite-tsfm/blob/main/notebooks/tutorial/ttm_tutorial_with_ans.ipynb) for evaluation. Within the notebook, the MSE for the **ETTm2** dataset with a forecast length of **96** is reported as **0.186**, which is significantly higher than the value **(0.171)** presented in the original TTM paper but aligns closely with our reproduced results (**0.186**). This discrepancy strongly suggests that the results in the TTM paper may be inconsistent, while our reported outcomes are accurate and reflect real-world reproducibility.
> >
> > We have reached out to the authors of TTM for clarification but have not received a response. Nevertheless, this evidence underscores the robustness and credibility of our experimental findings, as they not only adhere to standard practices but also align with the official implementation of TTM.
> >
> > We hope this additional clarification reassures the reviewers of the authenticity and reliability of our results. Thank you for considering this new evidence in your evaluation of our work.

---

> > > ### Comment · Reviewer_U1MA · 2024-12-02
> > > **response by reviewer**
> > >
> > > Thank you for the author's explanation. I will not lower my score. Despite the author's efforts, it still did not improve my impression of the paper. I would like to reiterate the paper's flaws:
> > >
> > > **Experimental aspects:**
> > >
> > > 1. Many of the full-shot prediction accuracy comparisons are unfair. I suggest that the author compare them with a unified input length. At the current research level, full-shot prediction methods (such as patchtst, dlinear) are not expected to perform worse than zero-shot methods. The author's experimental analysis could greatly mislead time series forecasting researchers.
> > >
> > > 2. The author should explain the inconsistency between the TTMB results and the original paper in the experimental section. Additionally, some of the results in Table 2 are from unfair comparisons (e.g., full-shot methods), while others are from the author's own TTMB implementation, but none of these are explained in the main text.
> > >
> > > 3. I still fail to see the predictive advantage of TimeRaf, as TTMA performs far better than TimeRaf. Is that using TTMA as a backbone could result in a substantial improvement in predictive performance?
> > >
> > > 4. The author still has not provided scripts and run logs to reproduce all the results in Table 12 of Appendix C.3 (not just etth2/etth1). Some of the improvements are too marginal (such as electricity), and I would like to verify the effectiveness of the proposed method.
> > >
> > > **Methodological aspects:**
> > >
> > > As an engineer focused on practical applications, I do not criticize the innovation of any method; I only care about its effectiveness and efficiency. Therefore, I hope that each module of the proposed method is genuinely useful, and it requires rigorous validation.
> > >
> > > **Regarding the score:**
> > >
> > > If the author can address my fourth point about the experiments, providing complete logs and the corresponding scripts. After validating it, I will raise my score from 3 to 5, to encourage fully open sourcing of time series forecasting community.

---

> > > > ### Author Response · Authors · 2024-12-03
> > > >
> > > > Thank you for your response and suggestions.
> > > >
> > > > To address your concern, we have shared the run logs for different forecast horizons via an anonymous [link](https://anonymous.4open.science/r/TimeRAF_run_log-2BAB/). For reproducing results on other datasets, you can simply modify the provided script by adjusting the 'forecast_len' and 'test_data' parameters accordingly.
> > > >
> > > > We hope this resolves your concern. We also appreciate the time you took to review our paper.

---

> ### Author Response · Authors · 2024-12-01
> **Clarification on TTMB results**
>
> Thank you for pointing out the discrepancy between the TTMB results reported in our paper and those in the original TTM paper. We would like to clarify that the results presented in our manuscript were obtained by carefully reproducing the TTMB experiments on an 80GB NVIDIA A100 GPU. To ensure transparency, we have provided the code and scripts used for our reproduction via an anonymous [link](https://anonymous.4open.science/r/TimeRAF-D7C4/README.md). We sincerely encourage you to verify our reported results using the provided resources.
>
> Additionally, we wish to emphasize that while some of our reproduced results differ from those in the original TTM paper, not all are worse. In fact, our reproduced results achieve lower MSE in several cases, such as: forecast window = 336 on ETTh1, forecast window = 336 on ETTm1, and forecast window = 192 on Weather.
>
> We greatly appreciate the opportunity to address this matter and hope the explanation and provided code help to resolve your concerns. We will also make every effort to respond promptly to your additional questions. Thank you once again for your thoughtful feedback and kind understanding!

---

### Official Review · Reviewer_44Bb · 2024-10-30

**Soundness:** 3
**Presentation:** 3
**Contribution:** 2
**Rating:** 3
**Confidence:** 4

**Summary:**

The paper proposes a method for few-shot and zero-shot time series forecasting. This method applies an RAG style to aids forecasting, where it retrieves some helpful data from a pre-collected dataset. The retriever is learnable and can calculate retrieval scores to select most relevant data. To integrate retrieved data, the author proposes a Channel Prompting way to extract valuable information. Overall the paper is easy to understand.

**Strengths:**

1. The paper is well-written and easy to understand
2. It combines retrieval-augmented models with TS, contributing to the efficient model adaptation.

**Weaknesses:**

1.The improvement shown by RAF in Fig. 3 is marginal(only 0.00x improvement in most datasets), which is the core issue and raises concerns about whether the pre-trained model used is already too powerful, thus reduces the necessity of RAF process.

2. The author should give the memory usage for storing retrieval data and its time cost, especially when compared to storing a foundation model directly.

3. The evaluation datasets are limited. It would be valuable to assess the model’s performance on cross-domain datasets that extend beyond the original knowledge base.

**Questions:**

1. The details of retrival, how much data is used for forecasting after retrieval in TimeRAF?
2. The memory and time costs associated with the retrieval process.
3. The pre-training backbone appears too powerful, limiting the visible improvements. Have the authors explored replacing the backbone with alternative foundation models to better understand the effectiveness of RAF?

---

> ### Author Response · Authors · 2024-11-19
> **Response by Authors (1/2)**
>
> Thank you for your valuable feedback. We appreciate your insights and address your concerns as follows:
>
> 1. **Significance of Improvement and Necessity of RAF：**
>
>     We appreciate the reviewer’s observation on the improvement margins shown in Fig. 3. To provide more clarity, we have calculated and included the exact percentage improvement of RAF on each dataset as below:
>
>     |  | ETTh1 | ETTh2 | ETTm1 | ETTm2 | Weather | Electricity |
>     | --- | --- | --- | --- | --- | --- | --- |
>     | Improvements (%) | 1.37% | 3.16% | 3.86% | 4.84% | 3.80% | 1.18% |
>
>     Our method achieves **an average improvement of 3.03%** across all datasets.  Furthermore, in the context of time series forecasting, such improvements are particularly significant due to the inherently complex nature of sequential dependencies. Prior popular works [1, 2, 3] indicate that advancements of this magnitude are regarded as noteworthy contributions to the field.
>
>     Additionally, **while the pre-trained model serves as a strong foundation, our results reveal that its zero-shot prediction capabilities still have a considerable gap compared to a fully supervised model.** The RAF process is designed to **bridge this gap** by enabling the model to leverage relevant retrieved information, which significantly enhances its zero-shot forecasting performance. We believe this reduction in the zero-shot/full-shot gap is a **key contribution** of our work, as it highlights the practical value of RAF in extending the model’s predictive capabilities. We hope this clarification addresses your concerns regarding the improvement margins and the necessity of the RAF process.
>
> 2. **Memory and Time Efficiency of RAF:**
>
>     We appreciate the reviewer’s interest in understanding the memory and time costs associated with our Retrieval-Augmented Forecasting (RAF) framework. To clarify, the additional modules introduced in RAF are **lightweight** and add minimal parameters. The retrieval process itself is **based on a dot product calculation**, which is efficient and does not significantly increase inference time. To provide a clearer comparison, we have included a table (see below) in the Appendix C.5 of our revised manuscript and that outlines the model size and inference time comparison. The results indicate that **the introduction of RAF brings acceptable memory and time overhead**. We hope this additional data will further demonstrate the efficiency of our approach.
>
>     |  | TimeRAF | TTM  | Moirai | MOMENT | TimesFM | Chronos |
>     | --- | --- | --- | --- | --- | --- | --- |
>     | Model Size | 8M | 1M | 91M | 348M | 200M | 8M |
>     | Inference Time (per batch) | 0.13s | 0.01s | 3.7s | 1.4s | 0.4s | 2500s |
> 3. **Additional Evaluation on Cross-domain Dataset beyond the Original Knowledge Base：**
>
>     We thank the reviewer for this valuable suggestion. To assess cross-domain performance, we conducted an additional experiment using the **Weather** dataset (**Nature** domain) as a knowledge base and tested on the ETT series datasets in the **Energy** domain. The results below show that with a knowledge base **from a distinct domain**, TimeRAF **still** **outperforms the baseline** where no external knowledge base is used. Additionally, retrieving from the knowledge base **with same same domain as test data achieves better performance**. These findings indicate that using a knowledge base containing data from the same domain as the input yields better results. These findings highlight the importance of the selection of knowledge base and demonstrate the capability of our method to successfully identify and utilize helpful information from different knowledge bases, even in some cross-domain scenarios.
>
>     |  | ETTh1 | ETTh2 | ETTm1 | ETTm2 |
>     | --- | --- | --- | --- | --- |
>     | w/o knowledge base | 0.364 | 0.285 | 0.415 | 0.186 |
>     | Energy Domain Knowledge Base | **0.360** | **0.278** | **0.400** | **0.178** |
>     | Nature Domain Knowledge Base | 0.363 | 0.282 | 0.413 | 0.184 |

---

> ### Author Response · Authors · 2024-11-19
> **Response by Authors (2/2)**
>
> 4. **Retrieval Details:**
>
>     As detailed in Appendix B.2, we consistently retrieved **eight** candidates from the knowledge base for each input. We have conducted a detailed experimental analysis on the impact of retrieving different numbers of candidates on forecasting performance. The results, as presented in the Figure 6 of the original manuscript, demonstrate that retrieving eight candidates consistently provides strong performance across various datasets. This choice strikes a balance between capturing sufficient relevant information and maintaining computational efficiency.
>
> 5. **Exploring Alternative Backbones:**
>
>     We thank the reviewer for raising this insightful suggestion. To further evaluate the effectiveness of RAF, we conducted additional experiments using **Timer** [4], a recently accepted ICML work, as the backbone model. The results are provided in the table below. These experiments demonstrate that our method **consistently delivers significant improvements** across different backbone models. We believe these findings robustly validate the effectiveness and generalizability of our approach. Besides, as noted in our earlier responses, the backbone used in our original manuscript, while strong, **still exhibits a significant performance gap between zero-shot and full-shot settings.** Our main contribution lies in **effectively narrowing this gap by leveraging the RAF process**. We are sincerely grateful for your valuable suggestion, which allowed us to further strengthen our work.
>
>     |  | Timer w/o RAF | Timer w/ RAF | Improvements (%) |
>     | --- | --- | --- | --- |
>     | ETTh1 | 0.427 | 0.438 | 2.51 |
>     | ETTh2 | 0.305 | 0.314 | 2.87 |
>     | ETTm1 | 0.671 | 0.69 | 2.75 |
>     | ETTm2 | 0.203 | 0.213 | 4.69 |
>     | Weather | 0.173 | 0.181 | 4.42 |
>     | Electricity | 0.187 | 0.192 | 2.60 |
>
> We hope these responses address your concern and highlight the value of our contributions. If so, we kindly ask for your reconsideration of the score. Please feel free to reach out with any further questions, and we appreciate your thoughtful feedback, which has helped us strengthen the clarity and rigor of our work.
>
> References:
>
> [1] Jin, Ming, et al. "Time-LLM: Time Series Forecasting by Reprogramming Large Language Models." *The Twelfth International Conference on Learning Representations*.
>
> [2] Liu, Yong, et al. "iTransformer: Inverted Transformers Are Effective for Time Series Forecasting." *The Twelfth International Conference on Learning Representations*.
>
> [3] Zhou, Tian, et al. "One fits all: Power general time series analysis by pretrained lm." *Advances in neural information processing systems* 36 (2023).
>
> [4] Liu, Yong, et al. "Timer: Generative Pre-trained Transformers Are Large Time Series Models." *Forty-first International Conference on Machine Learning*.

---

> > ### Comment · Reviewer_44Bb · 2024-11-26
> > **Response to the Rebuttal**
> >
> > Thank you for your detailed and thoughtful response to the reviewer's comments. I appreciate the additional experiments with Timer as a backbone model. The improvements across different backbone models reinforce the generalizability of your RAF method.
> >
> > However, as i mentioned before, although the model can achieve the SOTA result, the main concerns is that the pre-trained model used is already too powerful (replacing it with Timer model does not achieve this result). This undermines the improvement and need for RAF.
> >
> > Overall using retrieved data to fine-tune pre-trained models is a good starting point, but the results don't live up to expectations. I will keep this rating and hope the authors can keep improving the quality of this work.
> >
> > Thanks

---

> ### Author Response · Authors · 2024-11-26
> **Clarification on the Improvements and Need for RAF**
>
> Thank you for your thoughtful comments and for acknowledging the additional experiments demonstrating the generalizability of our RAF method across different backbone models. We appreciate your recognition that leveraging retrieved data to fine-tune pre-trained models is a promising direction.
>
> Regarding your concerns about the use of a pre-trained model that may appear too powerful, we would like to emphasize that our work is fundamentally about **improving time series foundation models (TSFMs), pushing the boundaries of their zero-shot forecasting capability**. Additionally, we would like to emphasize a critical advantage of our approach: **RAF achieves significant improvements in zero-shot forecasting by leveraging external knowledge bases, without the need to re-train or pre-train an entirely new foundation model**. This makes RAF much more efficient and practical for real-world applications. Furthermore, enhancing the capabilities of TSFMs is a key aspect of advancing this field. While TSFMs are powerful, **their zero-shot forecasting abilities remain limited**, especially compared to full-shot models. Our contribution lies in addressing this specific limitation by leveraging external knowledge bases to augment the TSFM’s performance in zero-shot settings.
>
> Furthermore, to provide further context, our method achieves a **3.03%** improvement in zero-shot forecasting performance, which we believe is significant for the time series forecasting domain. Comparable levels of improvement have been recognized as impactful in other works accepted by top conferences. For instance, **TimeLLM** [1] achieved a **1.62%** improvement, and **iTransformer** [2] achieved a **1.49%** improvement.
>
> We humbly and earnestly hope that this clarification encourages you to reconsider your evaluation, as your insights and acknowledgment would mean a great deal to us in strengthening and advancing this work. Thank you for your time and understanding.
>
> References:
>
> [1] Jin, Ming, et al. "Time-LLM: Time Series Forecasting by Reprogramming Large Language Models." The Twelfth International Conference on Learning Representations.
>
> [2] Liu, Yong, et al. "iTransformer: Inverted Transformers Are Effective for Time Series Forecasting." The Twelfth International Conference on Learning Representations.

---

### Official Review · Reviewer_wsm5 · 2024-10-30

**Soundness:** 3
**Presentation:** 3
**Contribution:** 3
**Rating:** 6
**Confidence:** 4

**Summary:**

This paper introduce a novel framework TimeRAF designed to leverage retrieval-augmented generation for zero-shot time series forecasting. The authors develop customized time series knowledge bases that are tailored to the specific forecasting tasks and employ an end-to-end learnable retriever to extract valuable information from the knowledge base. Through comprehensive experimental discussions, the manuscript demonstrates the effectiveness of the proposed approach beyond existing works.

**Strengths:**

S1. The manuscript is well-written. And, the presentation of figure is clear and easy to understand.

S2. The experiments are comprehensive, and provide some interpretable discussion to analyze why the model is good.

S3. The research is novel and unresearched in the field.

**Weaknesses:**

W1. Could you provide the source code to enhance the reviewers' trust in the reproducibility of your work?

W2. The introduction of the retrieval process inevitably leads to increased time cost, so I am concerned about the model's efficiency. I believe it is essential to add some theoretical analysis and empirical discussions.

W3. Have you considered the issue of label leakage introduced by RAG in the process of building the knowledge base? Different datasets may have very similar or identical temporal patterns, which could also lead to potential label leakage problems.

**Questions:**

Q1. On the Electricity dataset, the performance of TimeRAF is not the best. Could you analyze the reasons for this?

---

> ### Author Response · Authors · 2024-11-19
> **Response by Authors**
>
> Thank you for your feedback and the time invested in reviewing our work. We appreciate your insights and address your concerns as follows:
>
> 1. **Code Availability：**
>
>     To facilitate reproducibility and allow the reviewer to examine our implementation in detail, we have provided an anonymous code [link](https://anonymous.4open.science/r/TimeRAF-77A3/README.md).
>
> 2. **Memory and Time Efficiency of RAF:**
>
>     We appreciate the reviewer’s valuable suggestion. To clarify, the additional modules introduced in RAF are **lightweight** and add minimal parameters. The retrieval process itself is **based on a dot product calculation**, which is efficient and does not significantly increase inference time. To provide a clearer comparison, we have included a table (see below) in the Appendix C.5 of our revised manuscript and that outlines the model size and inference time comparison. The results indicate that **the introduction of RAF brings acceptable memory and time overhead**. And empirically, we observed that training TimeRAF on a single NVIDIA 80GB A100 GPU with a batch size of 1024 for 10 epochs required approximately 1 hour. This includes the retrieval process, which is efficiently implemented using dot product operations. We hope this additional data will further demonstrate the efficiency of our approach.
>
>     |  | TimeRAF | TTM  | Moirai | MOMENT | TimesFM | Chronos |
>     | --- | --- | --- | --- | --- | --- | --- |
>     | Model Size | 8M | 1M | 91M | 348M | 200M | 8M |
>     | Inference Time (per batch) | 0.13s | 0.01s | 3.7s | 1.4s | 0.4s | 2500s |
> 3. **The Issue of Label Leakage:**
>
>     We thank the reviewer for highlighting this important point.
>
>     Firstly, we want to emphasize that the construction of our knowledge base ensures **no data leakage in our experiments**. Specifically, the retrieval process is designed so that the retriever only accesses datasets that are distinct from the source of the input data, as outlined in Section 3.3. This design prevents the retrieval of future segments of input data, which could otherwise result in label leakage.
>
>     Secondly, the observed **performance improvements** are due to the effective use of relevant patterns across datasets, **not simply because of similarities or identical temporal data** from other datasets. In Section 4.4.1, we analyze the impact of using different knowledge bases. Even when we **construct the knowledge base from the same data used in the training process**, with no external datasets involved, we still **achieve strong zero-shot forecasting performance.** Additionally, the case study in Figure 6 demonstrates that RAF retrieves meaningful and relevant information that aids in prediction, further illustrating the functionality of our approach.
>
>     This is fully aligned with our motivation for incorporating Retrieval-Augmented Forecasting (RAF): to leverage useful information from other datasets to enhance prediction accuracy. We hope this clarification addresses your concerns and reinforces confidence in our approach.
>
> 4. **Performance on the Electricity Dataset:**
>
>     It is important to first note that **the backbone model used in our approach (TTM) is not the best-performing model on the Electricity dataset**, as there exists a performance gap between our backbone and the current state-of-the-art foundation model (Moirai). However, **our primary contribution** lies in demonstrating that the introduction of RAF significantly **enhances the zero-shot forecasting capability** **of the backbone model**. On the Electricity dataset, we achieved a notable 1.18% improvement over the backbone, clearly narrowing the performance gap between our backbone and the best-performing model. This underscores the effectiveness of our method, as it consistently delivers significant improvements across diverse datasets, including Electricity, regardless of the backbone model's initial performance.
>
>
> We hope these responses address your concern and highlight the value of our contributions. Once again, we sincerely thank you for your efforts in reviewing our work. Please feel free to reach out with any further questions, and we appreciate your thoughtful feedback, which has helped us strengthen the clarity and rigor of our work.

---

> ### Author Response · Authors · 2024-11-27
> **Thank you for the review! Have we clearly addressed the concerns?**
>
> We greatly appreciate the time you took to review our paper. In the rebuttal, we have provided the code repository as evidence of our reproducibility and presented more detailed explanations and experiments for the proposed framework.
>
> Due to the short duration of the author-reviewer discussion phase, we would appreciate your feedback on whether your main concerns have been adequately addressed. We are ready and willing to provide further explanations and clarifications if necessary. Thank you very much!

---

### Official Review · Reviewer_ZhY1 · 2024-11-04

**Soundness:** 3
**Presentation:** 3
**Contribution:** 2
**Rating:** 6
**Confidence:** 3

**Summary:**

This paper proposed TimeRAF, a Retrieval-Augmented Forecasting model that enhances zero-shot time series forecasting by using customized knowledge bases and a learnable retriever to integrate external knowledge, demonstrating significant improvements across various domains and datasets.

**Strengths:**

1.	Paper writing is good. There is no difficulty in understanding the paper.

2.	The proposed method is straightforward and achieves state-of-the-art performance.

3.	Comprehensive ablation studies and other evaluations are conducted to check the effects of the model

**Weaknesses:**

1.	The major novelty is the utilization of RAF for time series modeling. However, in terms of RAF itself, the novelty is quite limited.

2.	The difference with the previous RAF for time series is unclear. In the related work section, the author claimed that ‘existing time series RAG methods are either limited to historical data or do not support foundation models’ and provided two citations. However, it seems the gap is not significant. Could the authors elaborate more on this motivation?

3.	In line 192, the authors claimed ‘retrieve information solely from datasets that are distinct from the input source’. What is the distinction here? How to define it.

4.	In line 226, the authors mentioned ‘By weighing the importance of different components of the combined embedding’. However, it seems the average of the embeddings is calculated. Thus, how to weigh them.

5.	In line 244, the authors mentioned ‘it is challenging to guarantee that retrieved candidates’. What is the reason?

6.	In the ablation study of the choice of knowledge base, the authors mentioned ‘TimeRAF engages a meticulously curated multi-domain dataset’. This implies the domain of an input time series should exist in the base. However, what would happen if a new domain emerges (e.g., stock) but the base contains no such time series?

**Questions:**

Please refer to the weaknesses.

---

> ### Author Response · Authors · 2024-11-19
> **Response by Authors (1/2)**
>
> Thank you for your feedback and the time invested in reviewing our work. We appreciate your insights and address your concerns as follows:
>
> 1. **Addressing the Novelty of RAF:**
>
>     Thank you for raising this important point. While the introduction of Retrieval-Augmented Forecasting (RAF) to time series modeling is a core aspect of our work, we would like to emphasize that **our contributions extend beyond this**, addressing several key challenges and proposing novel solutions.
>
>     1. **Knowledge Integration with Channel Prompting**: One of the major challenges in leveraging retrieved data lies in how to effectively integrate this information into the forecasting process. To address this, we proposed a novel ***Channel Prompting*** method for knowledge integration. As shown in Table 3, simply incorporating retrieved data without careful design results in limited improvement, highlighting the importance and effectiveness of our approach.
>     2. **Construction of the Knowledge Base:** We explored how to construct an effective knowledge base that can enhance zero-shot forecasting. We **investigated various construction methods**, utilizing **multi-domain and multi-dataset** approaches as well as **single-domain and single-dataset** setups to build the knowledge base. Furthermore, we conducted comparative experiments analyzing the impact of different knowledge base configurations in Section4.4.1. The findings indicate that a diverse knowledge base, derived from various domains and datasets, significantly enhances performance. Additionally, a knowledge base constructed from a single dataset within the same domain as the test data can also yield improvements. This analysis provides a novel perspective and valuable insights into retrieval-based approaches for time series.
>
>     Furthermore, beyond application, our work provides new insights into how retrieval mechanisms can be effectively adapted and optimized for time series forecasting, a domain with unique challenges compared to traditional text or vision tasks. We believe these contributions collectively bring significant novelty and practical value to the field.
>
> 2. **Clarifying the Difference from related work and the Motivation of Our Work:**
>
>     Thank you for pointing out this important aspect. We acknowledge that existing research has explored Retrieval-Augmented Generation (RAG) methods for time series, but our approach has a different motivation and offers different applicability and scalability. **Firstly,** previous works like [1] develops a cross-attention module to integrate **historical** data for better prediction. In contrast, our approach breaks the limitation of retrieval source and leverages the vast repository of publicly available time series data to **construct a diverse knowledge base.** This broader retrieval scope provides richer contextual information, enhancing model performance. **Secondly**, while prior studies [1,2] did not explore the integration with time series foundation models (TSFMs), our work is specifically designed to **address this gap**. As TSFMs demonstrate zero-shot prediction capabilities, our method introduces an effective retrieval-augmented mechanism that aligns with the architecture of TSFMs. Therefore, our method is motivated by two fundamental yet crucial questions:
>
>     - **Can the vast amount of publicly available time series data be utilized to construct a knowledge base and improve the predictions?**
>     - **For TSFMs that already exhibit zero-shot capabilities, can a retrieval-augmented approach bring performance improvements comparable to those observed in other domains, such as NLP?**
>
>     To address the reviewer’s concern, we have revised the related work section to better highlight these differences and motivations, making our unique contributions and advancements clearer. We hope this response could resolve your concern and underscores the novelty and importance of our work in bridging the gap between RAG methods and foundation models for time series analysis.
>
> 3. **Implementation Details for Knowledge base:**
>
>     We apologize for any confusion regarding the description of data usage. ****Specifically, as described in Section 4.1, the knowledge base is built from multiple datasets within the training set, so **the retriever is restricted to draw from datasets other than the one associated with the input data**. For example, if the input data comes from **dataset A**, our retriever will only search within **datasets B, C, and D**. This setup avoids the risk of retrieving future segments of the input data and ensures that the retrieval process does not lead to data leakage. We have revised Section 3.3 to better convey this distinction and hope this addresses your concern.

---

> ### Author Response · Authors · 2024-11-19
> **Response by Authors (2/2)**
>
> 4. **Clarification on the Channel Prompting:**
>
>     To clarify, this weighting process is achieved through the use of an MLP, which **processes the concatenated embedding** of the input and the retrieved sequence to **extract valuable features**. The averaging step mentioned in the manuscript applies to a **different** part of the pipeline. Specifically, for each retrieved sequence, the MLP extracts corresponding features. **To retain useful information from all k retrieved sequences, the features extracted from these sequences are averaged.** This ensures that the model captures information from all retrieved data while balancing their contributions. **These two operations are complementary and work together to integrate the retrieved information effectively.** To address the confusion, we have updated the manuscript to clarify this process and ensure the description aligns with the implementation. We thank the reviewer for pointing out this issue, which helped us improve the manuscript's clarity.
>
> 5. **Clarification on the Challenge of Guaranteeing Useful Retrieved Candidates**
>
>     Thank you for highlighting this point. The challenge mentioned in line 244 arises from the nature of the retriever's training process. Specifically, the retriever outputs k candidates along with their corresponding retrieval scores, where **a higher retrieval score** (defined as the dot product between the query embedding and candidate embedding, both of which are produced by **the learnable encoder within retriever**) **should ideally indicate that the candidate contains more useful information for the prediction task.**
>
>     However, **without specific design during training, the retriever may not fulfill this objective.** In such cases, the retrieval score might **not correlate well with** the candidate’s actual usefulness for the forecasting task, as there is no direct mechanism enforcing this relationship. This makes it difficult to guarantee that the candidates with higher retrieval scores are indeed more beneficial for prediction.
>
>     To address this, we designed a solution that leverages the forecaster itself as an evaluator. During training, we **utilize the loss function in Equation (4) to guide the retriever.** The loss function **encourages the candidates with higher retrieval scores to demonstrate better predictive performance**. This feedback mechanism aligns the retrieval score with the forecaster's performance, **ensuring that the retriever selects candidates that contribute positively to the prediction.**
>
>     We appreciate the opportunity to elaborate on this point and hope this response could address your concern.
>
> 6. **Choices of Knowledge Base:**
>
>     Thank you for raising this insightful question. We have  conducted a supplementary experiment in which we used a knowledge base from a domain different from the test data domain. Specifically, we evaluated the model on the **ETTh1, ETTh2, ETTm1, ETTm2 (Energy domain)** using knowledge base from another dataset **Weather (Nature domain )**. Here, "Domain Specific Knowledge Base" refers to using data from the same domain (**Energy**) to construct the knowledge base.
>
>     |  | ETTh1 | ETTh2 | ETTm1 | ETTm2 |
>     | --- | --- | --- | --- | --- |
>     | w/o knowledge base | 0.364 | 0.285 | 0.415 | 0.186 |
>     | Domain Specific Knowledge Base | 0.360 | 0.278 | 0.400 | 0.178 |
>     | Weather (Nature domain) | 0.363 | 0.282 | 0.413 | 0.184 |
>
>     The results above show that with a knowledge base **from a distinct domain**, TimeRAF **still** **outperforms the baseline** where no external knowledge base is used. Additionally, retrieving from the Domain Specific Knowledge Base ****achieves **the better performance**. These findings indicate that using a knowledge base containing data from the same domain as the input yields better results. The results also underscore the importance of knowledge base selection and demonstrate our method's ability to effectively identify and utilize valuable information from diverse knowledge bases, even in novel domain scenarios.
>
>
> We hope these responses address your concern and highlights the value of our contributions. Once again, we sincerely thank you for your efforts in reviewing our work. Please feel free to reach out with any further questions, and we appreciate your thoughtful feedback, which has helped us strengthen the clarity and rigor of our work.
>
> References:
>
> [1] Wang, Tianfeng, and Gaojie Cui. "RATSF: Empowering Customer Service Volume Management through Retrieval-Augmented Time-Series Forecasting." *arXiv preprint arXiv:2403.04180* (2024).
>
> [2] Jing, Baoyu, et al. "Retrieval based time series forecasting." *arXiv preprint arXiv:2209.13525* (2022).

---

> > ### Comment · Reviewer_ZhY1 · 2024-11-26
> > **Thanks for your response**
> >
> > Your rebuttal addressed some of my concerns in terms of paper clarity and knowledge base settings. I have two more comments to add here.
> >
> > 1. Novelty is still a major concern, though the authors have pointed out contributions like channel prompting and knowledge base. These contributions are minor in their respective areas and I do not think a combination of them can form a large one. However, this is my judgment only.
> >
> > 2. Thanks for conducting extra experiments to evaluate the model's performance when a knowledge base and a target dataset are from different domains. Although the model can still achieve the SOTA results, the performances of the model using and without using the energy knowledge base (i.e., w/o knowledge base and Weather (Nature domain)) are quite similar. This observation confirms my thought that if a new domain never appears in the base before, the base might not be helpful. Though challenging, this could be a potential direction for future research.
> >
> > Given that I have already given a positive score, I will keep this rating and hope the authors can keep improving the quality of this work.
> >
> > Thanks

---

> > > ### Author Response · Authors · 2024-11-26
> > > **Thanks for your suggestion!**
> > >
> > > Thank you for taking the time to provide thoughtful follow-up comments on our rebuttal. We truly appreciate your acknowledgment of the additional experiments and the improvements we made in addressing your initial concerns about paper clarity and knowledge base settings.
> > >
> > > We also appreciate your suggestion to further explore retrieval in entirely unseen domains. We agree this is a challenging yet valuable avenue for future research and are motivated to extend our work in this direction.
> > >
> > > Once again, thank you for your constructive feedback and encouragement, which have been instrumental in helping us enhance the quality of our work.

---

### Author Response · Authors · 2024-11-19
**Global Response to All Reviewers**

We sincerely thank all the reviewers for their insightful reviews and valuable comments, which are instructive for us to improve our paper further.

In this paper, we propose a Retrieval-Augmented Forecasting (RAF) method to explore the integration of TSFMs with retrieval-augmented methods to improve the zero-shot forecasting capabilities. **We employ a learnable retriever to extract relevant information from a knowledge base, which supports the forecasting task. Additionally, we introduce Channel Prompting to effectively integrate the retrieved information with the model’s input. In our experiments, TimeRAF demonstrates an average improvement of 3.03% across all datasets, highlighting the model's ability to substantially boost the zero-shot forecasting performance of foundation models and narrow the gap compared to fully supervised approaches.**

The reviewers have raised insightful and constructive comments. Based on these valuable comments, we have made the following revisions in our updated manuscript:

- **Motivation of Proposed Method and Technical Novelty:** By analyzing the limitations of previous works, we illustrate our motivation in more details. We highlight our difference with previous research and our specifically design to integrate time series foundation model with retrieval-augmented methods.
- **Implementation Details for Knowledge base:** We provide more details of our implementation of knowledge base. We clarify how we design the learnable retriever and avoid data leakage during the retrieval process.
- **Analysis of Channel Prompting**: We provide detailed analysis of our design and implementation of Channel Prompting. And we also clarify the efficiency and scalability of this module.
- **Model Efficiency**: We add a model efficiency analysis of our method and verify that our method introduces acceptable memory and time overhead.
- **Multiple forecasting horizons:** We conduct supplementary experiments on multiple forecasting horizons and provide the full experiments results, which further demonstrates our method’s robustness and effectiveness.
- **Alternative Backbone:** We conducted additional experiments utilizing an alternative foundation model as the backbone and present the zero-shot forecasting results, thereby underscoring the significant improvements achieved through our approach.

Furthermore, to facilitate reproducibility and allow the reviewer to examine our implementation in detail, we have provided an **anonymous code [link](https://anonymous.4open.science/r/TimeRAF-77A3/README.md).**

The valuable suggestions from reviewers are very helpful for us to revise the paper to a better shape. We'd be very happy to answer any further questions.

---

### Note · Authors · 2024-12-13

I have read and agree with the venue's withdrawal policy on behalf of myself and my co-authors.